# TREE-BASED ACTION-MANIPULATION ATTACK AGAINST CONTINUOUS REINFORCEMENT LEARNING WITH PROVABLY EFFICIENT SUPPORT

## ABSTRACT

Due to the widespread application of reinforcement learning, research on its adversarial attacks is necessary for building secure reinforcement learning applications. However, most of the current security research focuses only on reinforcement learning with discrete states and actions, and these methods cannot be directly applied to reinforcement learning in continuous state and action spaces. In this paper, we investigate attacks on continuous reinforcement learning. Rather than manipulating rewards, observations, or environments, our focus lies in action-manipulation attacks that impose more restrictions on the attacker. Our study investigates the action-manipulation attack in both white-box and black-box scenarios. We propose a black-box attack method called LCBT, which uses a layered binary tree structure-based refinement and segmentation method to handle continuous action spaces. Additionally, we prove that under the condition of a sublinear relationship between the dynamic regret and total step counts of the reinforcement learning agent, LCBT can force the agent to frequently take actions according to specified policies with only sublinear attack cost. We conduct experiments to evaluate the effectiveness of the LCBT attack on three widely-used reinforcement learning algorithms: DDPG, PPO, and TD3.

## 1 INTRODUCTION

Reinforcement learning (RL) aims to maximize the accumulated rewards through the interaction between the agent and the environment. With the increasing complexity of scenarios, in many cases, reinforcement learning algorithms for discrete state-action environments are no longer applicable, while the application of continuous reinforcement learning is becoming more and more widespread in various domains, such as robot control (Xu et al., 2020), autonomous driving (Elallid et al., 2022), game intelligence (Jeerige et al., 2019) and etc. However, these scenarios demand high levels of security, thus, investigating its security problems such as adversarial attacks is crucial (García & Fernández, 2015).

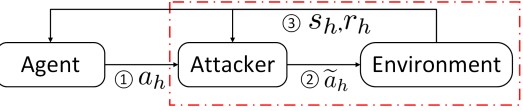

Figure 1: *Action-manipulation attack model. Step 1, the intelligent agent selects action $a_h$ based on its own policy and is intercepted by the attacker. Step 2, the attacker manipulates action $a_h$ to $\widetilde{a}_h$ based on their own strategy and submits it to the environment. Step 3, the environment responds to $\widetilde{a}_h$, and the attacker and the intelligent agent receive the current state and reward.*

In reinforcement learning, the interaction information between the agent and the environment is crucial for training. Therefore, manipulating observations, rewards, actions, and the environment can all have an impact on the training process. Attackers can exploit these different attack methods to influence the agent and achieve their own goals. In this paper, we focus on action manipulation. Specifically, the attacker operates as a third party between the agent and the environment, accessing interaction data such as state $s$, reward $r$, and action $a$. In this type of attack, the attacker achieves the attack by manipulating the actions generated by the intelligent agent into other actions in the action space. For example, in autonomous driving or robotics applications, the attacker can hijack

and tamper with the action signals. The attack process is illustrated in Fig. 1. It is evident that compared to manipulating observations, rewards, or the environment, action-manipulation is not as direct and efficient. Therefore, under such conditions, attackers need to intricately design their attack strategies to achieve their objectives.

Under the condition of manipulating actions for attack, the attacker can only manipulate within the agent's action space. Therefore, assessing actions and selecting appropriate action substitutions for the original actions are crucial for achieving efficient attacks. Based on action-manipulation attack, (Liu & Lai, 2021) proposed an attack algorithm that can force the agent to learn a specified policy while providing an upper bound on the attack cost. Prior to each tampering attempt, the algorithm evaluates all actions in the action space and then determines which action to select as a replacement for the original action. However, this method in paper (Liu & Lai, 2021) is clearly not applicable in a continuous action space. The attack strategy proposed in (Sun et al., 2020) also allows attackers to achieve an attack by manipulating actions. Its principle involves training an adversarial critic network using RL trajectories and then manipulating the interaction information between the agent and the environment to mislead the agent. Although the algorithm in (Sun et al., 2020) supports continuous action spaces, the attack cost bound is not given, and it requires knowledge of the RL algorithm used by the agent.

This study aims to explore methods for enabling an intelligent agent to learn specified policies through action-manipulation while minimizing the associated attack cost. Our research includes investigations in both white-box and black-box scenarios. We develop corresponding attack algorithms and establish bounds on the attack cost. In the white-box attack scenario, the attacker possesses comprehensive knowledge of the involved Markov decision processes (MDPs), enabling a more intuitive evaluation of actions and the design of attack strategies. Conversely, in the black-box attack scenario, the attacker lacks awareness of the underlying MDP and solely relies on information extracted from the RL trajectory to devise attack strategies. Additionally, in the black-box setting, a binary tree-based dynamic partitioning method is employed to thoroughly assess the action space. This method ensures the continual partitioning and refinement of the action space and achieves action evaluation through the utilization of importance sampling and probability-based mathematical methods. Our main contributions are as follows:

- We have constructed a threat model for action-manipulation attacks under continuous state and action spaces, considering the attacker's goal, knowledge, and capability. To adapt to the continuous action space, we propose the concept of target action space.
- In the white-box scenario, we utilize our understanding of the underlying MDP to intuitively propose the oracle attack method. We show that the oracle attack can force the agent who runs a sub-linear-regret RL algorithm to choose actions according to the target policies with sublinear attack cost.
- In the black-box scenario, we introduce an attack method called Lower Confidence Bound Tree (LCBT). It can be demonstrated that the effectiveness of this attack method closely approximates the oracle attack.
- We employ the proposed attack methods to target three popular reinforcement learning algorithms: DDPG (Lillicrap et al., 2015), PPO (Schulman et al., 2017), and TD3 (Fujimoto et al., 2018). The experimental results demonstrate the effectiveness of our proposed methods.

## 2 RELATED WORK

We elucidate the work related to the security of reinforcement learning from the perspectives of different attack methods.

**Reward manipulation:** (Ma et al., 2019), (Zhang et al., 2020), (Ma et al., 2018) and (Huang & Zhu, 2019) employ reward manipulation to train the agent to learn a specified policy, while (Majadas et al., 2021) focuses on minimizing the total rewards of the agent using this attack approach. In the study by (Wu et al., 2022), the modification of rewards was implemented in a multi-agent system to facilitate the learning of several agents' target policies.

**Observation manipulation:** (Foley et al., 2022) employs observation manipulation to cause agent misbehavior only at specific target states. (Zhang et al., 2021), (Yang et al., 2020), (Pan et al., 2019) minimize the total rewards of the intelligent agent using the observation manipulation method.

Moreover, (Behzadan & Munir, 2017) utilizes the method of observation manipulation to achieve policy induction attacks.

**Environment manipulation:** (Xu et al., 2021) train agents to use a target policy for specific observations through environment manipulation. (Tanev et al., 2021) and (Huang et al., 2017) minimize the total rewards of the agent through environment manipulation. In (Boloor et al., 2019), the environment is manipulated to reach a target state.

**Action manipulation:** In RL with discrete state and action spaces, (Liu & Lai, 2021) manipulates actions to force the intelligent agent to learn the specified policy. (Lee et al., 2020) minimizes the overall reward of the intelligent agent through action manipulation. In addition, (Tessler et al., 2019) focuses on action robust RL in the presence of noisy environments.

## 3 PRELIMINARY

This paper considers a finite-horizon MDP over the continuous domains. Such an MDP can be defined as a 6-tuple $M = (S, A, P, r, H, \mu)$. where $S$ and $A$ are respectively bounded continuous state and action spaces, and $H$ denotes the number of steps per episode. $\mu$ is the initial state distribution. $P$ is the transition matrix so that $P_h(\cdot|s, a)$ gives the distribution over states if action $a$ is taken for state $s$ at step $h \in [H]$, and $r_h : S \times A \to [0, 1]$ is the deterministic reward function at step $h$. Define $K$ is the total number of episodes. For each episode $k \in [K]$, through the interaction between the agent and the environment over $H$ steps, the trajectory $\{s_1, a_1, r_1, s_2, a_2, r_2, ..., s_H, a_H, r_H\}$ can be obtained. This trajectory provides essential data for further training.

We represent the agent's Markov policy by $\pi$. Here, $\pi_h : S \to A$ depicts the policy at the step $h$, while $\pi_h(\cdot|s_h)$ portrays the probability distribution of actions selected by policy $\pi_h$, corresponding to the state $s_h$. A deterministic policy is a policy that maps each state to a particular action. For notation convenience, for a deterministic policy $\pi$, we use $\pi_h(s)$ to denote the action $a$ which satisfies $\pi_h(a|s) = 1$.

A policy $\pi$ can be evaluated by the expected reward. Formally, We use the $Q_h^\pi : S \times A \to R$ to express the expected value obtained by selecting action $a_h$ at the state $s_h$ according to the policy $\pi$, which is defined as $Q_h^\pi(s, a) := r_h(s, a) + \mathbb{E}\left[\sum_{h'=h+1}^{H} r_{h'}(s_{h'}, \pi_{h'}(s_{h'}))|s_h = s, a_h = a\right]$.

Accordingly, we define $V_h^\pi : S \to R$ to represent the total expected rewards in state $s_h$ under policy $\pi$, and it is denoted by $V_h^\pi(s, a) := \mathbb{E}_\pi\left[\sum_{h'=h}^{H} r_{h'}(s_{h'}, \pi_{h'}(s_{h'}))|s_h = s\right]$. For a finite-horizon MDP, we set $V_{H+1}^\pi = 0$ and $Q_{H+1}^\pi = 0$.

The purpose of reinforcement learning is to find the optimal policy $\pi^*$, which gives the optimal value function $V_h^*(s) = \sup_\pi V_h^\pi(s)$. To assess a reinforcement learning algorithm's performance over $K$ episodes, we quantify a metric known as regret, which is defined as

$$\text{Regret}(K) = \sum_{k=1}^{K} \left[V_1^*(s_1^k) - V_1^{\pi_k}(s_1^k)\right]. \tag{1}$$

We now state an additional geometrical definition for the state and action spaces.

**Definition 1.** *The state space $S$ is equipped with a dissimilarity function $l_s : S^2 \to \mathbb{R}$ such that $l_s(s, s') \geq 0$ for all $(s, s') \in S^2$ and $l_s(s, s) = 0$. Likewise, The action space $A$ is equipped with a dissimilarity function $l_a : A^2 \to \mathbb{R}$ such that $l_a(a, a') \geq 0$ for all $(a, a') \in A^2$ and $l_a(a, a) = 0$.*

For a subset $D \subseteq S$, the diameter of it is defined as $diam_s(D) := \sup_{x,y \in D} l_s(x, y)$. Likewise, we define $diam_a(P) := \sup_{x,y \in P} l_a(x, y)$ where $P \subseteq A$.

## 4 ATTACK STRATEGY AND ANALYSIS

Firstly, we introduce the attack model. In each step $h \in [H]$ at episode $k \in [K]$, the attacker has to decide whether to take action-manipulation method, i.e., launch attack when receives the action $a$ generated by the agent, and $\widetilde{a} \in A$ is the potential action taken by the attacker. If the attacker

decides not to attack, $\widetilde{a} = a$. Also, the attacker has a deterministic target policy $\pi^\dagger$ and a radius $r_a$. Define $\mathcal{A}_h^\dagger(s) = \{a : l_a(a, \pi_h^\dagger(s)) \leq r_a\}$ as the target action space.

We describe the threat model in this paper from the perspective of attacker's goal, attacker's knowledge, and attacker's capability.

- **Attacker's Goal:** Let $\tau$ be the step set whose element is the step when the attacker launches attack, i.e., $\tau = \{(k, h) : a_h^k \neq \widetilde{a}_h^k, k \in [1, K], h \in [1, H]\}$. And let $\alpha$ be the step set whose element is the step when the action $a_h^k$ meets the condition $a_h^k \notin \mathcal{A}_h^\dagger(s_h^k)$, i.e., $\alpha = \{(k, h) : a_h^k \notin \mathcal{A}_h^\dagger(s_h^k), k \in [1, K], h \in [1, H]\}$. The attacker's goal is to manipulate the agent into picking its actions in the target action space, in other words, following the policies similar to the target policy $\pi^\dagger$. At the same time, the attacker aims to minimize both the cost $|\tau|$ and loss $|\alpha|$.
- **Attacker's Knowledge:** In the context of the black-box attack, the attacker has no prior knowledge about the underlying environment and the agent's policy. It only has access to the interaction information between the agent and the environment, i.e., $s_h^k$, $a_h^k$, and $r_h^k$.
- **Attacker's Capability:** The attacker can manipulate the action $a_h^k$ generated by the agent and change it to another action $\widetilde{a}_h^k$. In addition, the attacker has a certain amount of computational resources and storage space to run the attack algorithms.

## 4.1 ORACLE ATTACK

In the white-box attack scenario, the attacker has the full information of the underlying MDP $\mathcal{M}$. Therefore, the attacker can compute the $Q$-values and $V$-values of any policy.

Define a policy set

$$\Pi^\dagger = \{\pi : \pi_h(s) \in \mathcal{A}_h^\dagger(s), V_h^\pi(s) \geq V_h^\dagger(s), \forall s \in S, h \in [H]\}, \tag{2}$$

which represents a set of policies that produce actions within the target action space and are superior to policy $\pi^\dagger$, and let $\pi^o = \sup_{\pi \in \Pi^\dagger} V_h^\pi(s)$. For notation simplicity, we denote $V_h^\dagger(s) := V_h^{\pi^\dagger}(s)$ and $V_h^o(s) := V_h^{\pi^o}(s)$. We define $a_h^-(s) = \arg\min_{a \in A} Q_h^\dagger(s, a)$ as the worst action for a given target policy $\pi^\dagger$ and state $s$ in the step $h$.

Based on the above, a possible attack approach intuitively is that the attacker can mislead the agent by manipulating actions to make it believe that policies in the set $\Pi^\dagger$ are better than policies that generate actions outside the target action space. We now introduce an effective oracle attack strategy. Specifically, at the step $h$ and state $s$, if the action selected by the agent is within the target action space, i.e., $a_h \in \mathcal{A}_h^\dagger(s)$, the attacker does not launch an attack, i.e., $\widetilde{a}_h = a_h$. Otherwise, i.e., $a_h \notin \mathcal{A}_h^\dagger(s)$, the attacker launches attack and sets $\widetilde{a}_h = a_h^-(s)$. It should be noted that if the target policy is the worst policy, there may potentially exist $V_h^o(s) = V_h^\dagger(s) = \inf_\pi V_h^\pi(s)$ for the state $s$ and step $h$, in this case, the attacker can not mislead the agent to learn the policies in the set $\Pi^\dagger$. To ensure the feasibility of the attack, the target policy $\pi^\dagger$ needs to satisfy the following condition

$$V_h^\dagger(s) > Q_h^\dagger(s, a_h^-(s)), \forall s \in S, h \in [H], \tag{3}$$

which means that $\pi^\dagger$ is not the worst policy. Define the minimum gap $\Delta_{min}^\dagger$ by $\Delta_{min}^\dagger = \min_{h \in [H], s \in S} \left(V_h^\dagger(s) - Q_h^\dagger(s, a_h^-(s))\right)$. Under (3), $\Delta_{min}^\dagger > 0$ also holds. Then, we have:

**Lemma 1.** *When $\Delta_{min}^\dagger > 0$ holds, in the observation of the agent, the policies in set $\Pi^\dagger$ are superior to other policies that do not align with the attacker's target.*
*Proof. Please refer to the supplementary.*

With the condition $\Delta_{min}^\dagger > 0$, the upper bounds of $\tau$ and $\alpha$ under the oracle attack can be obtained.

**Theorem 1.** *With a probability at least $1 - \delta_2$, the oracle attack will force the agent to learn the policies in $\Pi^\dagger$ with $\tau$ and $\alpha$ bounded by*

$$|\tau| \leq |\alpha| \leq \frac{\text{Regret}(K) + 2H^2\sqrt{\ln(1/\delta_2) \cdot \text{Regret}(K)}}{\Delta_\dagger^o + \Delta_{min}^\dagger}, \tag{4}$$

*where $\Delta_\dagger^o = \min_{h \in [H], s \in S} \left(V_h^o(s) - V_h^\dagger(s)\right)$.*
*Proof: Please refer to the supplementary.*

---

**Algorithm 1** LCBT attack strategy on continuous RL algorithm

---

**Input:**

Target policy $\pi^\dagger$.

Initialize $\mathcal{T}_1^h = \{(0,1),(1,1),(1,2)\}$, $\hat{Q}_{1,1}^h(1) = \hat{Q}_{1,2}^h(1) = 0$, $L_{1,1}^h(1) = L_{1,2}^h(1) = -\infty$, and $T_{1,1}^h(1) = T_{1,2}^h(1) = 0$, for all $h \in [1, H]$.

**for** episode $k = 1, 2, \ldots, K$ **do**

    Receive $s_1^k$. Initialize the set of trajectory $traj = \{s_1^k\}$.

    **for** step $h = 1, 2, \ldots, H$ **do**

        The agent chooses an action $a_h^k$.

        **if** $a_h^k \in \mathcal{A}_h^\dagger(s_h^k)$ **then**

            The attacker does not attack, i.e., $\widetilde{a}_h^k = a_h^k$, and sets $w_h = 1$.

        **else**

            Take $\{(D_k^h, I_k^h), P_k^h\} \leftarrow$ WorTraverse$(\mathcal{T}_k^h)$, and set $\widetilde{a}_h^k = a_{D_k^h, I_k^h}$, $w_h = 0$.

        **end if**

        The environment receives action $\widetilde{a}_h^k$, and returns the reward $r_h^k$ and the next state $s_{h+1}^k$.

        Update the trajectory by plugging $\widetilde{a}_h^k$, $r_h^k$ and $s_{h+1}^k$ into $traj$.

    **end for**

    Set the cumulative reward $G_{H+1:H+1} = 0$ and the importance ratio $\rho_{H+1:H+1} = 1$.

    **for** step $h = H, H-1, \ldots, 1$ **do**

        **if** $\widetilde{a}_h^k \in \mathcal{A}_h^\dagger(s_h^k)$ **then**

            Continue

        **end if**

        $G_{h:H+1} = r_h^k + G_{h+1:H+1}$, $\rho_{h:H+1} = \rho_{h+1:H+1} \cdot w_h$, $t_k^h = T_{D_k^h, I_k^h}^h(k) \leftarrow T_{D_k^h, I_k^h}^h(k) + 1$.

        Use Eq.(5) to update the value $\hat{Q}_{D_k^h, I_k^h}^h(k)$

        **if** $\nu_1 \rho^{D_k^h} \geq \frac{H-h+1}{\sqrt{2t_k^h}} \sqrt{\ln\left(\frac{2Mk\sum_{h=1}^H |\mathcal{T}_k^h|}{\delta_1}\right)}$ AND $(D_k^h, I_k^h) \in$ leaf$(\mathcal{T}_k^h)$ **then**

            $\mathcal{T}_k^h \leftarrow \mathcal{T}_k^h \cup \{(D_k^h+1, 2I_k^h-1), (D_k^h+1, 2I_k^h)\}$

            $L_{D_k^h+1, 2I_k^h-1}(k) = L_{D_k^h+1, 2I_k^h}(k) = -\infty$

        **end if**

        **for all** $(D, I) \in \mathcal{T}_k^h$ **do**

            Use Eq.(7) to update the value $L_{D,I}^h(k)$;

            Use Eq.(8) to update the value $B_{D,I}^h(k)$ backward from leaf nodes.

        **end for**

    **end for**

**end for**

---

We can see that the oracle attack relies on the knowledge of $a_h^-(s)$, which hinders the direct application of it to many real-world scenarios. In the next subsection, we will introduce the black-box attack method.

## 4.2 LCBT ATTACK

In the black-box attack scenario, the attacker has no knowledge of the underlying MDP process and the algorithm used by the agent, and only knows the interaction information: $s_h^k$, $a_h^k$, and $r_h^k$. Without the knowledge of $a_h^-(s)$, the attacker can not determine $\widetilde{a}$ and fails to attack. To overcome this issue, the attacker can use a method to estimate $a_h^-(s)$. Specifically, the attacker utilizes the hierarchical binary tree structure to discretize the action space, with the state as the context to evaluate $Q_h^\dagger(s,a)$, and then construct an attack algorithm that approximates the oracle attack. The attack algorithm has been designated as *LCBT* (Lower Confidence Bound Tree), in relation to its utilization of a tree structure.

**Action Cover Tree:** As the action space $A$ is continuous, we use a binary tree $\mathcal{T}$ to discretize the action space and reduce the possible options for selection. In the cover tree, we denote by $(D, I)$ the node at depth $D \geq 0$ with index $I \in [1, 2^D]$ among the nodes at the same depth. Clearly, the root

node is $(0, 1)$. The two children nodes of $(D, I)$ are denoted by $(D + 1, 2I - 1)$ and $(D + 1, 2I)$ respectively. For each node $(D, I)$, a continuous subset $\mathcal{P}_{D,I} \in A$ of actions is divided from the action space and associated with this node. The $\mathcal{P}_{D,I}$ can be determined recursively as $\mathcal{P}_{0,1} = A$, $\mathcal{P}_{D,I} = \mathcal{P}_{D+1,2I-1} \cup \mathcal{P}_{D+1,2I}$, and $\mathcal{P}_{D,I} \cap \mathcal{P}_{D,J} = \emptyset$. For each node $(D, I)$, an action $a_{D,I}$ is selected to represent the node. Whenever $(D, I)$ is sampled, action $a_{D,I}$ is selected. Moreover, the nodes in $\mathcal{T}$ need also satisfy the following assumption,

**Assumption 1.** *We assume that there exist constants $\nu_1$ and $0 < \rho < 1$, such that for each node $(D, I)$: $diam_a(\mathcal{P}_{D,I}) \leq \nu_1 \rho^D$.*

In addition, until the beginning of episode $k$, an action coverage tree is maintained for each step $h \in [H]$ and it is denoted as $\mathcal{T}_k^h$, and $|\mathcal{T}_k^h|$ is the node number.

**State Partition:** In order to better evaluate Q-values, we also need to discretize the continuous state space. Here, we partition it into $M$ subintervals $S_1, S_2, ..., S_{M-1}, S_M$ which satisfy the following conditions: $S = \cup_{i=1}^{M} S_i$, and $S_i \cap S_j = \emptyset$ for $\forall i, j \in [1, M]$. To ensure the functionality of the LCBT algorithm, the state division must satisfy the following assumption, avoiding rough partitions.

**Assumption 2.** *There exist constants $L_s$ and $d_s$ which satisfy $L_s d_s < \Delta_\dagger^o + \Delta_{min}^\dagger$ such that for all $S_i, \forall i \in [1, M]$: $diam_s(S_i) \leq L_s d_s$.*

Define $i(s) : s \in S_{i(s)}$, which represents the number of the subinterval to which state $s$ belongs.

**LCB Calculation:** LCB is the pessimistic estimate of unknown $Q_h^\dagger(s, a)$. In the black-box attack, at the step $h$ in episode $k$, the attacker will select a node $(D, I)$ and set $\tilde{a}_h^k$ as $a_{D,I}$ when $a_h^k \notin \mathcal{A}_h^\dagger(s_h^k)$. In the LCBT attack algorithm, for each node $(D, I)$, $\hat{Q}_{D,I}^h(s_h^k, a_{D,I})$ is used to evaluate $Q_h^\dagger(s_h^k, a_{D,I})$, which is calculated by

$$\hat{Q}_{D,I}^h(s_h^k, a_{D,I}) = (1 - \frac{1}{T_{D,I}^h(k)})\hat{Q}_{D,I}^h(s_h^{\gamma_{D,I}^h(k)}, a_{D,I}) + \frac{1}{T_{D,I}^h(k)}(r_h^k + G_{h+1:H+1}^k \cdot \rho_{h+1:H+1}^k), \quad (5)$$

where $T_{D,I}^h(k) = |\phi_{D,I}^h(k)|$, and $\phi_{D,I}^h(k) = \{\gamma : \gamma < k, s_h^\gamma \in S_{i(s_h^k)}, a_h^\gamma = a_{D,I}\}$, is defined as the set of episodes in which the current state belonged to subinterval $S_{i(s_h^k)}$ and node $(D, I)$ was selected by the attacker at the step $h$ until the beginning of the episode $k$. $\gamma_{D,I}^h(k) = \max\{\gamma : \gamma \in \phi_{D,I}^h(k)\}$ represents the latest episode before $k$ in which the current state belonged to the subinterval $S_{i(s_h^k)}$ and node $(D, I)$ was selected. $G_{h:H}^k = \sum_{h'=h}^{H} r_h^k$ is the cumulative reward. The importance sampling ratio is calculated by $\rho_{h:H}^k = \prod_{h'=h}^{H} \frac{\mathbb{P}(\tilde{a}_{h'}^k | s_{h'}^k, \pi_{h'}^\dagger)}{\mathbb{P}(\tilde{a}_{h'}^k | s_{h'}^k, b_{h'}^k)}$, where $b^k$ is the behavior policy that generates trajectory $\{s_1^k, \tilde{a}_1^k, r_1^k, s_2^k, \tilde{a}_2^k, r_2^k, ..., s_{H-1}^k, \tilde{a}_{H-1}^k, r_{H-1}^k, s_H^k, \tilde{a}_H^k, r_H^k, s_{H+1}^k\}$ and $\mathbb{P}(\tilde{a}_h^k | s_h^k, b_h^k)$ is,

$$\mathbb{P}(\tilde{a}_h^k | s_h^k, b_h^k) = \begin{cases} 1 & \text{if } \tilde{a}_h^k = a_h^k \text{ and } a_h^k \in \mathcal{A}_h^\dagger(s_h^k), \\ 1 & \text{if } \tilde{a}_h^k = a_{D_k^h, I_k^h} \text{ and } a_h^k \notin \mathcal{A}_h^\dagger(s_h^k), \end{cases} \quad (6)$$

otherwise, the value is $0$. Since we assume that the target policy is a deterministic function, we have $\mathbb{P}(\tilde{a}_h | s_h, \pi_h^\dagger) = \mathbb{I}\{\tilde{a} \in \mathcal{A}_h^\dagger(s)\}$. For the indicator function $\mathbb{I}\{\xi\}$, if event $\xi$ is established $\mathbb{I}\{\xi\} = 1$, otherwise $\mathbb{I}\{\xi\} = 0$. We set $\rho_{H+1:H+1}^k = 1$, $G_{H+1:H+1}^k = 0$ and $\rho_{h:H}^k = \rho_{h:H+1}^k$, $G_{h:H}^k = G_{h:H+1}^k$. According to the definition of $b^k$, we have $V_h^{b^k}(s) = \mathbb{E}[G_{h:H}^k | s_h^k = s]$ and $V_h^\dagger(s) = \mathbb{E}[\rho_{h:H}^k G_{h:H}^k | s_h^k = s]$.

The states used to calculate the $\hat{Q}_{D,I}^h(s, a_{D,I})$ value are limited to a subinterval, instead of being fixed. Also, each tree node $(D, I)$ covers the action space $\mathcal{P}_{D,I}$, despite being represented by action $a_{D,I}$. Thus, the lower confidence bound of node $(D, I)$ can be calculated by,

$$L_{D,I}^h(k) = \hat{Q}_{D,I}^h(s_h^k, a_{D,I}) - \frac{H - h + 1}{\sqrt{2T_{D,I}^h(k)}} \sqrt{\ln\left(\frac{2Mk \sum_{h=1}^{H} |\mathcal{T}_k^h|}{\delta_1}\right)} - L_s d_s - \nu_1 \rho^D. \quad (7)$$

We use coefficients $L_s d_s$ and $\nu_1 \rho^D$ to measure the level of uncertainty generated by the state subinterval and the node's action space, respectively. The second term is the confidence interval radius obtained by Hoeffding's inequality.

---

**Algorithm 2** The *WorTraverse* function.

---

**Input:**
$\mathcal{T}_k^h$
1: $(D, I) \leftarrow (0, 1), P \leftarrow (0, 1)$
2: **while** $(D, I) = (0, 1)$ OR $(D, I) \notin \text{leaf}(\mathcal{T}_k^h)$ **do**
3:    **if** $B_{D+1,2I-1}^h \leq B_{D+1,2I}^h$ **then**
4:       $(D, I) \leftarrow (D + 1, 2I - 1)$
5:    **else**
6:       $(D, I) \leftarrow (D + 1, 2I)$
7:    **end if**
8:    $P \leftarrow P \cup \{(D, I)\}$
9: **end while**
10: **return** $(D, I)$ and $P$

---

**Worst Node Selection:** When in the step $h$ of episode $k$, if the action $a_h^k$ chosen by the agent is not in $\mathcal{A}_h^{\dagger}(s_h^k)$, then the attacker will utilize the $L$-values to choose a node $(D, I)$, and use $a_{D,I}$ to replace $a_h^k$, that is, set $\tilde{a}_h^k$ to $a_{D,I}$. Furthermore, due to $\nu_1 \rho^{D+1} < \nu_1 \rho^D$, the child node has a smaller structural resolution, which reduces the uncertainty in the LCB estimate. Define the $B$-values,

$$B_{D,I}^h(k) = \begin{cases} L_{D,I}^h(k) & \text{if } (D, I) \in \text{leaf}(\mathcal{T}_k^h), \\ \max \left[ L_{D,I}^h(k), \min_{j \in \{2I-1, 2I\}} B_{D+1,j}^h(k) \right] & \text{otherwise.} \end{cases} \tag{8}$$

The B-values are designed to have a tighter lower bound on $Q_h^{\dagger}(s_h^k, a_{D,I})$ by taking the maximum between $L_{D,I}^h(k)$ for the current node, and the minimum lower bound of the node's two child nodes. Based on the B-values, the attacker traverses the tree $\mathcal{T}_k^h$ from the root node with smaller B-values to the leaf node, and the path is represented as $P_k^h$. The traverse function is shown in Algorithm 2.

After one episode, the attacker begins to utilize the interaction data between the agent and the environment to update the $\hat{Q}$, $L$, and $B$ values of nodes. A critical step is deciding when to expand a node into two child nodes to reduce the uncertainty that is caused by the size of the node. Intuitively, a node should be expanded when it is chosen a particular number of times such that the radius of the confidence interval is approximately equal to the node's size. This occurs when the uncertainty brought by the node's size begins to dominate. Therefore, for node $(D_k^h, I_k^h)$, the attacker expands it into its two child nodes when the following equation is true, i.e.,

$$\nu_1 \rho^{D_k^h} \geq \frac{H - h + 1}{\sqrt{2t_k^h}} \sqrt{\ln \left( \frac{2Mk \sum_{h=1}^H |\mathcal{T}_k^h|}{\delta_1} \right)}, \tag{9}$$

and set the $L$-values of the two child nodes as $-\infty$. Essentially, because the *WorTraverse* function selects nodes with smaller B-values, nodes containing $a_h^-(s_h^k)$ are more likely to be expanded, thereby reducing the uncertainty caused by node size and sufficiently approximating action $a_h^-(s_h^k)$.

## 4.3 MAIN RESULTS

Next, we present the lemmas and theorems concerning the LCBT attack in the black-box setting.

**Lemma 2.** *Under the LCBT attack, the following confidence bound:*

$$\left| \hat{Q}_{D,I}^h(s_h^k, a_{D,I}) - \mathbb{E}[\hat{Q}_{D,I}^h(s_h^k, a_{D,I})] \right| \leq \frac{H - h + 1}{\sqrt{2T_{D,I}^h(k)}} \sqrt{\ln \left( \frac{2Mk \sum_{h=1}^H |\mathcal{T}_k^h|}{\delta_1} \right)} \tag{10}$$

*holds for $\forall h \in [H], m \in [M], (D, I) \in \mathcal{T}_h^k, T_{D,I}^h(k) \in [1, k]$ with a probability at least $1 - \delta_1$. It should be noted that for $\mathbb{E}[\hat{Q}_{D,I}^h(s_h^k, a_{D,I})]$, there exists a state $s_{k,h}^0 \in S_{i(s_h^k)}$ such that $Q_h^{\dagger}(s_{k,h}^0, a_{D,I}) = \mathbb{E}[\hat{Q}_{D,I}^h(s_h^k, a_{D,I})]$ holds.*
*Proof: Please refer to the supplementary.*

The lemma offers a confidence bound for $\hat{Q}_{D,I}^h(s_h^k, a_{D,I})$. According to the Algorithm 2, the attacker traverses the tree with smaller B-values, which means $\hat{Q}_{D,I}^h(s_h^k, a_{D,I})$ will converge to $Q_h^\dagger(s_h^k, a_h^-(s_h^k))$. Lemma 3 gives the relationship between $\mathbb{E}[\hat{Q}_{D,I}^h(s_h^k, a_{D,I})]$ and $Q_h^\dagger(s_h^k, a_h^-(s_h^k))$.

**Lemma 3.** *In the LCBT algorithm, based on the lemma 2, there exists,*

$$\mathbb{E}[\hat{Q}_{D,I}^h(s_h^k, a_{D,I})] - Q_h^\dagger(s_h^k, a_h^-(s_h^k)) \leq 3 \cdot \frac{H - h + 1}{\sqrt{2T_{D,I}^h(k)}} \sqrt{\ln\left(\frac{2Mk\sum_{h=1}^H |\mathcal{T}_k^h|}{\delta_1}\right)} + L_s d_s. \quad (11)$$

*Proof: Please refer to the supplementary.*

Before presenting the main theorem, it is important to note that if the attacker and the environment are considered as a single entity, the resulting environment is non-stationary. Therefore, there may be different optimal policies corresponding to different episodes, especially at the beginning of the black-box attack. As a result, the measurement of regret also undergoes changes. In plain terms, dynamic regret measures the degree of regret resulting from comparing the policy adopted by an agent with the optimal policy of each specific episode in hindsight. On the other hand, static regret only compares the agent's policy with the optimal fixed policy derived from combining all episodes. According to Fei et al. (2020),the definition of the dynamic regret is D-Regret$(K) := \sum_{k\in[K]} \left[ V_1^{\pi^{*,k},k}(s_1^k) - V_1^{\pi^k,k}(s_1^k) \right]$, where $\pi^{*,k} = \sup_\pi V_1^{\pi,k}(s_1^k)$ is the optimal policy of episode $k$.

Our main theorem is the upper bound of cost $|\tau|$ and loss $|\alpha|$ in the context of using the LCBT attack algorithm.

**Theorem 2.** *With a probability at least $1 - \delta_1 - \delta_2$, the LCBT attack will force the agent to learn the policies in $\Pi^\dagger$ with $\tau$ and $\alpha$ bounded by*

$$|\tau| \leq |\alpha| \leq \frac{\text{D-Regret}(K) + 2H^2\sqrt{\ln(1/\delta_2) \cdot \text{D-Regret}(K)}}{\Delta_\dagger^o + \Delta_{min}^\dagger - L_s d_s} + \frac{18MH^2 \ln(\frac{2MHK^2}{\delta_1})\sum_{h=1}^H |\mathcal{T}_K^h|}{(\Delta_\dagger^o + \Delta_{min}^\dagger - L_s d_s)^2}, \quad (12)$$

*with $|\mathcal{T}_K^h| = O(K^E), E = \log_{2\rho^{-2}} 2 < 1$ for $\forall h \in [H]$.*
*Proof: Please refer to the supplementary.*

**Remark.** When D-Regret$(K) \leq \frac{MH^3 K^E \ln(2MHK^2/\delta_1)}{\Delta_\dagger^o + \Delta_{min}^\dagger - L_s d_s}$, we have the upper bound $\frac{MH^3 K^E \ln(2MHK^2/\delta_1)}{(\Delta_\dagger^o + \Delta_{min}^\dagger - L_s d_s)^2}$. In other words, a lower attack cost may be incurred when the reinforcement learning algorithm utilized by the agent works better in non-stationary environments. The denominator of $\Delta_\dagger^o + \Delta_{min}^\dagger - L_s d_s$ indicates that the attack cost is higher when the target policy approaches the worst policy or $r_a$ is smaller.

## 5 NUMERICAL RESULTS

In this section, we provide numerical experiments of our oracle attack and LCBT attack on the three popular continuous reinforcement learning algorithms, including DDPG, PPO, and TD3, in two different environments. The experiments are run on a machine with NVIDIA Quadro RTX 5000.

Environment 1 involves a one-dimensional continuous control problem with $s \in [-1, 1]$ and $a \in [-1, 1]$ where a slider moves on a rail, receiving positive rewards proportional to the move distance, with a negative reward when it falls off. Environment 2 describes a two-dimensional continuous control problem with $s \in [0, 8]^2$ and $a \in [-1, 1]^2$ where a vehicle moves on a two-dimensional plane, receiving linear rewards proportional to the distance from the central point for every step. The target policy $\pi^\dagger$ is trained by constraining the movement range of both the slider and vehicle. The oracle attack and LCBT attack algorithms are utilized to attack against the DDPG and PPO algorithms in Environment 1, while attacking against DDPG and TD3 algorithms in Environment 2. The results for each environment are depicted in Fig. 2 and Fig. 3.

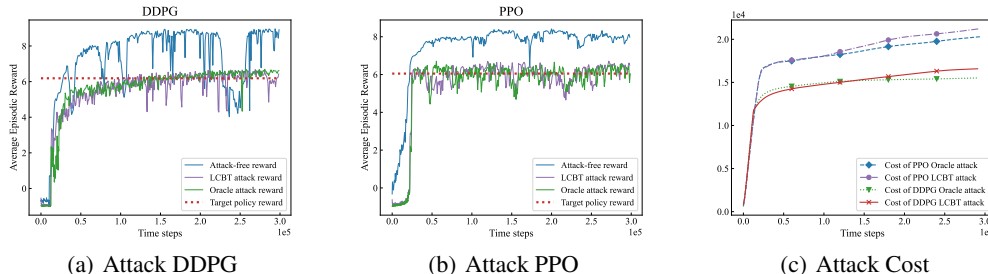

(a) Attack DDPG        (b) Attack PPO        (c) Attack Cost

Figure 2: *Reward and cost results of Environment* 1. *In this experiment, we set the similarity action radius* $r_a = 0.0625$, *the state sub-interval quantity* $M = 16$, *the number of steps per episode* $H = 10$, *and the total time steps* $T = 3 * 10^5$. $\rho$ *is set to be* $1/2$. *The x-axis represents the time step* $t$ *with the total time step. In (a, b), the y-axis represents the average reward of the last 10 episodes. In (c), the y-axis represents the cumulative cost that changes over time steps.*

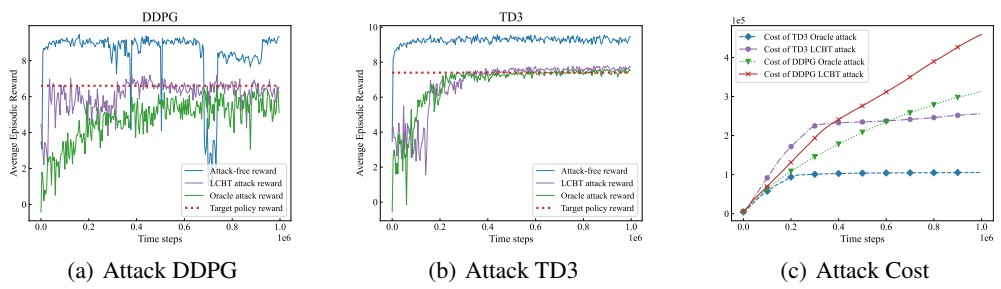

(a) Attack DDPG        (b) Attack TD3        (c) Attack Cost

Figure 3: *Reward and cost results of Environment* 2. *In this experiment, we set* $r_a = 0.31$, $M = 81$, $H = 10$, *and* $T = 10^6$. $\rho$ *is set to be* $1/\sqrt{2}$.

In Fig. 2(a, b) and Fig. 3(a, b), the target policy reward (- - -) is the average reward achieved by the $\pi^\dagger$ over $10^3$ episodes. The convergence of rewards obtained by the intelligent agent under three different conditions (attack-free, LCBT attack, and oracle attack) is represented by the remaining three curves. The experimental results reveal that the LCBT attack and oracle attack cause the agent to learn policies near the $\pi^\dagger$, this indicates the effectiveness of our attack algorithm. Fig. 2(c) and Fig. 3(c) show the cumulative cost of each attack algorithm as a function of the time steps. Specifically, Compared with the oracle attack, the LCBT attack based on the black-box setting requires more cost to achieve the attack target. Furthermore, the cost is sublinear to time steps in both the oracle attack and the LCBT attack, which is in line with our theoretical expectations.

In addition, it can be observed that the convergence speed of the algorithm in environment 2 is slower compared to environment 1. This is because environment 2 has higher complexity, requiring the attacker to take more steps to approach the worst action, thus increasing the non-stationarity of the environment. Furthermore, from Fig. 3(c), it can be seen that the corresponding attack cost of TD3 is smaller, indicating that TD3 algorithm has a stronger ability to deal with non-stationary environments, resulting in a smaller D-Regret. Due to the limited space, more results can be found in the appendix.

## 6 CONCLUSION

In this study, we investigated the action-manipulation attack to disrupt reinforcement learning in continuous state and action spaces. We defined the attack model based on the attacker's goal, knowledge, and capability. We investigated the impact of two attack models, the oracle and black-box attacks, that vary in the attacker's level of knowledge. Our theoretical analysis and empirical experiments validate the efficacy of the proposed black-box attack method in forcing the intelligent agent to follow the attacker's target policies at a sublinear cost. Additionally, the black-box attack LCBT can achieve a result approaching the oracle attack as runtime increases. In future research, we will develop a robust reinforcement learning algorithm to resist the examined attacks and enhance the defense mechanisms.

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

## APPENDIX

The organization of the appendix is as follows. Firstly, We will introduce the experimental environments in detail in Section A. Then we will show the proofs of Lemma 1 and Theorem 1 in Section B and C respectively. The proofs of Lemma 2 and Lemma 3 are represented in Section D and E, respectively. In section F, the proof of Theorem 2 is represented. In Section G, we present additional results. Finally, in Section H, we will have a discussion. Table 1 summarizes the notations used in our paper.

## A    EXPERIMENTAL SETTINGS

**Environment** 1. The objective of this environment, as depicted in Fig. 4, is to control the slider to slide on the rod and maximize the reward within $H = 10$ steps. Specifically, for any $h \in [H]$, the state space $s \in [-1, 1]$ represents the current position of the slider on the rod, where $0$ represents the center point of the rod. The action space $a \in [-1, 1]$ represents the base sliding distance, with negative values signifying leftward movement and positive values signifying rightward movement. The final sliding distance $d = a * 2$, and a reward of $|a|$ is obtained for each step taken. If the slider falls off the rod (i.e., the slider's position coordinate is outside of $[-1, 1]$), the round of the

Table 1: *Notation Table*

| Notation | Meaning |
|---|---|
| $S$ | The state space |
| $A$ | The action space |
| $H$ | The number of total steps in each episode |
| $\mu$ | The initial state distribution |
| $K$ | The number of total episodes |
| $s_h^k$ | The state at step $h$ in episode $k$ |
| $a_h^k$ | The action taken by the agent at step $h$ in episode $k$ |
| $\widetilde{a}_h^k$ | The action taken by the attacker and submitted to the environment at step $h$ in episode $k$ |
| $r_h^k$ | The reward generated at step $h$ in episode $k$ |
| $l_s$ | The dissimilarity function of the state space |
| $l_a$ | The dissimilarity function of the action space |
| $diam_s(D)$ | The largest difference of all the states in set $D \subseteq S$ w.r.t $l_s$ |
| $diam_a(P)$ | The largest difference of all the actions in set $P \subseteq A$ w.r.t $l_a$ |
| $\pi^\dagger$ | The target policy specified by the attacker |
| $b^k$ | The behavior policy of episode $k$ |
| $\pi_h^\dagger(s)$ | The target action for state $s$ at step $h$ |
| $r_a$ | The radius specified by the attacker |
| $\mathcal{A}_h^\dagger(s)$ | The target action space, which is defined as $\{a : l_a(a, \pi_h^\dagger(s)) \leq r_a\}$ |
| $\Pi^\dagger$ | The set of target policies that produce actions within the target action space and are superior to policy $\pi^\dagger$ |
| $\tau$ | The step set whose element is the step when the attacker launches the attack |
| $\alpha$ | The step set whose element is the step when action $a_h^k$ does not belong to the target action space $\mathcal{A}_h^\dagger(s_h^k)$ |
| $V_h^\dagger(s)$ | The value which is equivalent to $V_h^{\pi^\dagger}(s)$ |
| $Q_h^\dagger(s,a)$ | The value which is equivalent to $Q_h^{\pi^\dagger}(s,a)$ |
| $a_h^-(s)$ | The worst action for a given target policy $\pi^\dagger$ and state $s$ at step $h$ |
| $\Delta_\dagger^o$ | The minimum gap between the policies $\pi^o$ and $\pi^\dagger$ |
| $\Delta_{min}^\dagger$ | The minimum gap between the policy $\pi^\dagger$ and the worst action |
| $\mathcal{T}_k^h$ | The action cover tree that has been built by the attacker until the beginning of episode $k$ w.r.t step $h$ |
| $P_k^h$ | The traverse path at step $h$ in episode $k$ |
| $|\mathcal{T}_k^h|$ | The node number of tree $\mathcal{T}_k^h$ |
| $\nu_1$ | The maximum distance w.r.t. $l_a$ in the action space |
| $\rho$ | The reduction ratio of nodes in $\mathcal{T}_k^h$ at different depth |
| $(D, I)$ | The node at depth $D$ with index $I$ in action cover trees |
| $a_{D,I}$ | The represented action of node $(D, I)$ |
| $M$ | The number of total subintervals of the state space |
| $S_m$ | A subinterval of the state space with $m \in [1, M]$ |
| $i(s)$ | The number of the subinterval to which state $s$ belongs, i.e., $s \in S_{i(s)}$. |
| $d_s$ | The diameter of subintervals of the state space |
| $L_s$ | The coefficient of $d_s$ |
| $\phi_{D,I}^h(k)$ | The set of episodes in which the current state belonged to subinterval $S_{i(s_h^k)}$ and node $(D, I)$ was selected by the attacker at step $h$ until the beginning of the episode $k$ |
| $\gamma_{D,I}^h(k)$ | The latest episode before $k$ in which the current state belonged to the subinterval $S_{i(s_h^k)}$ and node $(D, I)$ was selected |
| $T_{D,I}^h(k)$ | The value is equal to $|\phi_{D,I}^h(k)|$ |
| $\hat{Q}_{D,I}^h(s_h^k, a_{D,I})$ | The calculated value used to evaluate $Q_h^\dagger(s_h^k, a_{D,I})$ |
| $L_{D,I}^h(k)$ | The lower confidence bound of node $(D, I)$ |
| $B_{D,I}^h(k)$ | The tighter lower confidence bound of node $(D, I)$ |

game ends immediately and a reward of $-1$ is received. The slider's initial position $s_1$ is arbitrarily set between $-0.7$ and $0.7$. Our target policy $\pi^\dagger$ for the slider is to move only within the interval $[-0.7, 0.7]$. We generate the target policy model by training and obtaining the optimal policy within the limited interval $[-0.7, 0.7]$ through augmentation of the environment constraints. The optimal policy is then employed to act as the target policy. Obviously, $\pi^\dagger$ is not the globally optimal policy.

**Environment** 2. As depicted in Fig. 5, the objective of this environment is to control a vehicle to move on a two-dimensional plane surrounded by a boundary and obtain maximum reward within $H = 10$ steps. For any $h \in [H]$, the state space of Environment 2 includes the current two-dimensional position of the vehicle, denoted as $\boldsymbol{s} = (s_1, s_2) \in [0, 8]^2$. The action space, denoted as $\boldsymbol{a} = (a_1, a_2) \in [-1, 1]^2$, denotes the distance of the vehicle's movement. The vehicle's next position is $(s_1 + a_1, s_2 + a_2)$ and the reward gained is linearly proportional to the distance $d$ between the vehicle and the center point (the yellow dot at $[4, 4]$). The closer the distance, the higher the reward. The initial position $s_1$ of the vehicle is at any point outside the red circle ($d > 1$). Our target policy $\pi^\dagger$ for the vehicle is to move only outside the red circle. Similarly, we train and obtain the optimal policy that only moves outside the red circle by increasing the environmental constraints, which serves as the target policy. Obviously, $\pi^\dagger$ is not the globally optimal policy.

**Environment** 3. Environment 3 is a five-dimensional version of environment 3, where the goal remains the same: to control an object to move as close as possible to the center point within $H = 10$ steps to obtain maximum reward. The state space is denoted as $\boldsymbol{s} = (s_1, s_2, s_3, s_4, s_5) \in [0, 8]^4$, the action space as $\boldsymbol{a} = (a_1, a_2, a_3, a_4, a_5) \in [-1, 1]^5$, and the next state is given by $(s_1 + a_1, s_2 + a_2, s_3 + a_3, s_4 + a_4, s_5 + a_5)$. The center point is located at $[4, 4, 4, 4, 4]$. In this environment, the Euclidean distance is used as the metric. The attacker's target policy $\pi^\dagger$ is defined as getting as close to the center point as possible without entering the range less than 1 unit distance from the center point.

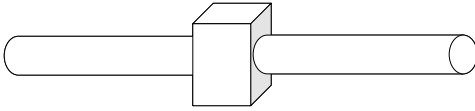

Figure 4: *Environment* 1. *The objective of this environment is to control the slider to slide on the rod*

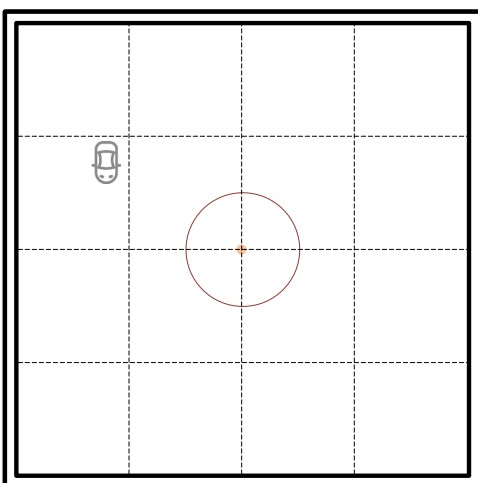

Figure 5: *Environment* 2. *The objective of this environment is to control a vehicle to move on a two-dimensional plane*

## B  THE PROOF OF LEMMA 1

In the action-manipulation settings, the attacker sits between the agent and the environment. We can regard the combination of the agent and the environment as a new environment. For the new environment, we represent the $Q$-value and $V$-value as $\overline{Q}$ and $\overline{V}$. Assume that the attacker selects actions according to the target policy $\pi^\dagger$, and the attacker will not launch the attack, then we have

$$\overline{Q}_h^\dagger(s, \pi_h^\dagger(s)) = Q_h^\dagger(s, \pi_h^\dagger(s)) = V_h^\dagger(s). \tag{13}$$

If the attacker selects an action $a \notin \mathcal{A}_h^\dagger(s)$, the attacker will change it to $a_h^-(s)$, then we have,

$$\overline{Q}_h^\dagger(s, \pi_h^-(s)) = Q_h^\dagger(s, a_h^-(s)) \overset{(i)}{<} Q_h^\dagger(s, \pi_h^\dagger(s))$$
$$= V_h^\dagger(s) \overset{(ii)}{=} \overline{Q}_h^\dagger(s, \pi_h^\dagger(s)), \forall \pi^- : \pi_h^-(s) \notin \mathcal{A}_h^\dagger(s), h \in [H],$$

where $(i)$ is because $\Delta_{min}^\dagger > 0$, $(ii)$ is according to (13). Then we have, i.e.,

$$\overline{Q}_h^\dagger(s, \pi_h^-(s)) < V_h^\dagger(s) \le V_h^\pi(s) \overset{(i)}{=} \overline{V}_h^\pi(s), \text{ for } \forall \pi \in \Pi^\dagger, \tag{14}$$

where $(i)$ is because when the agent follows the policies $\pi \in \Pi^\dagger$, the attacker will not launch the attack. The proof is completed.

## C  PROOF OF THEOREM 1

Define $\overline{\Delta}_h^k = \overline{V}_h^o(s_h^k) - \overline{Q}_h^o(s_h^k, a_h^k)$. From Lemma 1 and the definition of $\pi^0 = \sup_{\pi \in \Pi^\dagger} V_h^\pi(s), \forall s, h$, in the observation of the agent under the oracle attack, $\pi^o$ is the optimal policy, then the regret of the agent's performance can be defined as

$$\text{Regret}(K) = \sum_{k=1}^K [\overline{V}_1^*(s_1^k) - \overline{V}_1^{\pi^k}(s_1^k)]$$
$$= \sum_{k=1}^K [\overline{V}_1^o(s_1^k) - \overline{V}_1^{\pi^k}(s_1^k)], \tag{15}$$

where $\pi^k$ is the policy followed by the agent for each episode $k$.

For episode $k$,

$$\overline{V}_1^o(s_1^k) - \overline{V}_1^{\pi^k}(s_1^k)$$

$$= \overline{V}_1^o(s_1^k) - \mathbb{E}_{a \sim \pi_1^k(\cdot|s_1^k)}[\overline{Q}_1^o(s_1^k, a)] + \mathbb{E}_{a \sim \pi_1^k(\cdot|s_1^k)}[\overline{Q}_1^o(s_1^k, a)] - \overline{V}_1^{\pi^k}(s_1^k)$$

$$= \overline{\Delta}_1^k + r_1^k + \mathbb{E}_{s' \sim \mathbb{P}(\cdot|s_1^k, a \sim \pi_1^k(\cdot|s_1^k))} \overline{V}_2^o(s') - (r_1^k + \mathbb{E}_{s' \sim \mathbb{P}(\cdot|s_1^k, a \sim \pi_1^k(\cdot|s_1^k))} \overline{V}_2^{\pi^k}(s'))$$

$$= \overline{\Delta}_1^k + \mathbb{E}_{s' \sim \mathbb{P}(\cdot|s_1^k, a \sim \pi_1^k(\cdot|s_1^k))}[\overline{V}_2^o(s') - \overline{V}_2^{\pi^k}(s')]$$

$$= \overline{\Delta}_1^k + \mathbb{E}_{s' \sim \mathbb{P}(\cdot|s_1^k, a \sim \pi_1^k(\cdot|s_1^k))} \left[ \overline{V}_2^o(s') - \mathbb{E}_{s'' \sim \mathbb{P}(\cdot|s', a' \sim \pi_2^k(\cdot|s'))}[\overline{Q}_2^o(s', a')] + \mathbb{E}_{s'' \sim \mathbb{P}(\cdot|s', a' \sim \pi_2^k(\cdot|s'))}[\overline{Q}_2^o(s', a')] - \overline{V}_2^{\pi^k}(s') \right]$$

$$= \overline{\Delta}_1^k + \mathbb{E}_{s' \sim \mathbb{P}(\cdot|s_1^k, a \sim \pi_1^k(\cdot|s_1^k))}[\overline{\Delta}_2^k] +$$
$$\mathbb{E}_{s' \sim \mathbb{P}(\cdot|s_1^k, a \sim \pi_1^k(\cdot|s_1^k))} \left[ r_2^k + \mathbb{E}_{s'' \sim \mathbb{P}(\cdot|s', a' \sim \pi_2^k(\cdot|s'))} \overline{V}_3^o(s'') - r_2^k - \mathbb{E}_{s'' \sim \mathbb{P}(\cdot|s', a' \sim \pi_2^k(\cdot|s'))} \overline{V}_3^{\pi^k}(s'') \right]$$

$$= \overline{\Delta}_1^k + \mathbb{E}_{s' \sim \mathbb{P}(\cdot|s_1^k, a \sim \pi_1^k(\cdot|s_1^k))}[\overline{\Delta}_2^k] + \mathbb{E}_{s' \sim \mathbb{P}(\cdot|s_1^k, a \sim \pi_1^k(\cdot|s_1^k))} \left[ \mathbb{E}_{s'' \sim \mathbb{P}(\cdot|s', a' \sim \pi_2^k(\cdot|s'))}[\overline{V}_3^o(s'') - \overline{V}_3^{\pi^k}(s'')] \right]$$

$$= ... = \mathbb{E}[\sum_{h=1}^H \overline{\Delta}_h^k | \mathcal{F}_1^k]$$

where $\mathcal{F}_h^k$ represents the $\sigma$-field generated by all the random variables until episode $k$, step $h$ begins. So there exists

$$\sum_{k=1}^K \left( \overline{V}_1^o(s_1^k) - \overline{V}_1^{\pi^k}(s_1^k) \right) = \sum_{k=1}^K \mathbb{E}[\sum_{h=1}^H \overline{\Delta}_h^k | \mathcal{F}_1^k] = \mathbb{E}[\sum_{k=1}^K \sum_{h=1}^H \overline{\Delta}_h^k | \mathcal{F}_1^k]. \tag{16}$$

Next, we will show that with a probability of at least $1 - \delta_2$, we have

$$\sum_{k=1}^{K}\sum_{h=1}^{H}\overline{\Delta}_h^k \le \sum_{k=1}^{K}\left(\overline{V}_1^o(s_1^k) - \overline{V}_1^{\pi^k}(s_1^k)\right) + 2H^2\sqrt{\ln(1/\delta_2)\sum_{k=1}^{K}\left(\overline{V}_1^o(s_1^k) - \overline{V}_1^{\pi^k}(s_1^k)\right)}. \quad (17)$$

Since $\mathbb{E}[\sum_{h=1}^{H}\overline{\Delta}_h^k|\mathcal{F}_1^k] = \overline{V}_1^o(s_1^k) - \overline{V}_1^{\pi^k}(s_1^k)$, we can regard

$$Y_k := \sum_{i=1}^{k}\left(\sum_{h=1}^{H}\overline{\Delta}_h^k - \left(\overline{V}_1^o(s_1^k) - \overline{V}_1^{\pi^k}(s_1^k)\right)\right) \quad (18)$$

as a martingale with the difference sequence $\{X_k\}_{k=1}^{K}$, which is

$$X_k := \sum_{h=1}^{H}\overline{\Delta}_h^k - \left(\overline{V}_1^o(s_1^k) - \overline{V}_1^{\pi^k}(s_1^k)\right). \quad (19)$$

And we have the difference sequence bounded, i.e., $|X_k| \le H^2, \forall k \in [1, K]$. Define the predictable quadratic variation process of the martingale: $W_K := \sum_{k=1}^{K}\mathbb{E}[X_k^2|\mathcal{F}_1^k]$, with

$$W_K \le \sum_{k=1}^{K}\mathbb{E}\left[(\sum_{h=1}^{H}\overline{\Delta}_h^k)^2|\mathcal{F}_1^k\right] \le H^2\sum_{k=1}^{K}\mathbb{E}\left[\sum_{h=1}^{H}\overline{\Delta}_h^k|\mathcal{F}_1^k\right] = H^2\sum_{k=1}^{K}\left(\overline{V}_1^o(s_1^k) - \overline{V}_1^{\pi^k}(s_1^k)\right). \quad (20)$$

By Freeman's inequality Tropp (2011), we have

$$\mathbb{P}\left(\sum_{k=1}^{K}X_k > 2H^2\sqrt{\ln(1/\delta_2)\sum_{k=1}^{K}\left(\overline{V}_1^{\dagger}(s_1^k) - \overline{V}_1^{\pi^k}(s_1^k)\right)}\right)$$

$$\le \exp\left\{-\frac{4H^4\ln(1/\delta_2)\sum_{k=1}^{K}\left(\overline{V}_1^{\dagger}(s_1^k) - \overline{V}_1^{\pi^k}(s_1^k)\right)/2}{W_K + H^2 \cdot 2H^2\sqrt{\ln(1/\delta_2)\sum_{k=1}^{K}\left(\overline{V}_1^{\dagger}(s_1^k) - \overline{V}_1^{\pi^k}(s_1^k)\right)}/3}\right\}$$

$$\le \exp\{-\ln(1/\delta_2)\} = \delta_2.$$

Under the oracle attack, when the agent chooses an action satisfying $a_h \in \mathcal{A}_h^{\dagger}(s_h)$, the attacker does nothing, and we have $\overline{Q}_h^o(s_h^k, a_h^k) = Q_h^o(s_h^k, a_h^k) \le V_h^o(s_h^k)$. Otherwise, the attacker launches the attack, and the action $a_h^k$ will be replaced by $a_h^-(s_h^k)$. In other words, we have $\overline{Q}_h^o(s_h^k, a_h^k) = Q_h^{\dagger}(s_h^k, a_h^-(s_h^k))$. Then, we can obtain

$$\sum_{k=1}^{K}\sum_{h=1}^{H}\overline{\Delta}_h^k$$

$$= \sum_{k=1}^{K}\sum_{h=1}^{H}\left(\overline{V}_h^o(s_h^k) - \overline{Q}_h^o(s_h^k, a_h^k)\right) = \sum_{k=1}^{K}\sum_{h=1}^{H}\left(V_h^o(s_h^k) - \overline{Q}_h^o(s_h^k, a_h^k)\right)$$

$$= \sum_{(k,h)\notin\alpha}V_h^o(s_h^k) - Q_h^o(s_h^k, a_h^k) + \sum_{(k,h)\in\alpha}V_h^o(s_h^k) - Q_h^{\dagger}(s_h^k, a_h^-(s_h^k))$$

$$\ge \sum_{(k,h)\in\alpha}V_h^o(s_h^k) - Q_h^{\dagger}(s_h^k, a_h^-(s_h^k))$$

$$= \sum_{(k,h)\in\alpha}V_h^o(s_h^k) - V_h^{\dagger}(s_h^k) + V_h^{\dagger}(s_h^k) - Q_h^{\dagger}(s_h^k, a_h^-(s_h^k))$$

$$\ge \sum_{(k,h)\in\alpha}\Delta_{\dagger}^o + \Delta_{min}^{\dagger}$$

$$= |\alpha|(\Delta_{\dagger}^o + \Delta_{min}^{\dagger}).$$

Lastly, we have

$$|\alpha| \leq \frac{\sum_{k=1}^{K}\sum_{h=1}^{H}\overline{\Delta}_h^k}{\Delta_\dagger^o + \Delta_{min}^\dagger} \overset{(i)}{\leq} \frac{\text{Regret}(K) + 2H^2\sqrt{\ln(1/\delta_2)\cdot\text{Regret}(K)}}{\Delta_\dagger^o + \Delta_{min}^\dagger},$$

$(i)$ is obtained by (17). With $|\tau| \leq |\alpha|$, the proof is completed.

## D   PROOF OF LEMMA 2

Firstly, we will transform $\hat{Q}$-value into a non-recursive form, i.e.,

$$
\begin{aligned}
&\hat{Q}_{D,I}^h(s_h^k, a_{D,I}) \\
&= (1 - \frac{1}{T_{D,I}^h(k)})\hat{Q}_{D,I}^h(s_h^{\gamma_{D,I}^h(k)}, a_{D,I}) + \frac{1}{T_{D,I}^h(k)}(r_h^k + G_{h+1:H+1}^k \cdot \rho_{h+1:H+1}^k) \\
&= \frac{T_{D,I}^h(k)-1}{T_{D,I}^h(k)}\Big[(1 - \frac{1}{T_{D,I}^h(k)-1})\hat{Q}_{D,I}^h(s_h^{\gamma_{D,I}^h(\gamma_{D,I}^h(k))}, a_{D,I}) \\
&\qquad\qquad + \frac{1}{T_{D,I}^h(k)-1}(r_h^{\gamma_{D,I}^h(k)} + G_{h+1:H+1}^{\gamma_{D,I}^h(k)} \cdot \rho_{h+1:H+1}^{\gamma_{D,I}^h(k)})\Big] + \frac{1}{T_{D,I}^h(k)}(r_h^k + G_{h+1:H+1}^k \cdot \rho_{h+1:H+1}^k) \\
&= ... = \frac{1}{T_{D,I}^h(k)}\sum_{i=1}^{k}\mathbb{I}\{s_h^i \in S_{i(s_h^k)}, a_h^i = a_{D,I}\}(r_h^i + G_{h+1:H+1}^i \cdot \rho_{h+1:H+1}^i).
\end{aligned}
$$

Then we have

$$
\begin{aligned}
\mathbb{E}\left[\hat{Q}_{D,I}^h(s_h^k, a_{D,I})\right] &= \frac{1}{T_{D,I}^h(k)}\sum_{i=1}^{k}\mathbb{I}\{s_h^i \in S_{i(s_h^k)}, a_h^i = a_{D,I}\}\mathbb{E}\left[r_h^i + G_{h+1:H+1}^i \cdot \rho_{h+1:H+1}^i\right] \\
&= \frac{1}{T_{D,I}^h(k)}\sum_{i=1}^{k}\mathbb{I}\{s_h^i \in S_{i(s_h^k)}, a_h^i = a_{D,I}\}Q_h^\dagger(s_h^i, a_h^i).
\end{aligned}
$$

Define the event:

$$
\begin{aligned}
\xi_k = \big\{ &\forall T_{D,I}^h(k) \in [1,k], h \in [H], m \in [M], (D,I) \in \mathcal{T}_k^h, \\
&\left|\hat{Q}_{D,I}^h(s_h^k, a_{D,I}) - \mathbb{E}\left[\hat{Q}_{D,I}^h(s_h^k, a_{D,I})\right]\right| \leq \beta\left(k, T_{D,I}^h(k), \delta_1\right) \big\},
\end{aligned}
$$

where $\beta$-function is calculated by

$$\beta(k, N, \delta) = \frac{H-h+1}{\sqrt{2N}}\sqrt{\ln\left(\frac{2Mk\sum_{h=1}^{H}|\mathcal{T}_k^h|}{\delta}\right)}. \tag{21}$$

Define $\xi_k^c$ as the opposite event of $\xi_k$, then, we have

$$
\begin{aligned}
\mathbb{P}(\xi_k^c) &= \sum_{h=1}^{H} \sum_{m=1}^{M} \sum_{(D,I) \in \mathcal{T}_k^h} \sum_{T_{D,I}^h(k)=1}^{k} \mathbb{P}\left( \left| \hat{Q}_{D,I}^h(s_h^k, a_{D,I}) - \mathbb{E}\left[ \hat{Q}_{D,I}^h(s_h^k, a_{D,I}) \right] \right| > \beta(k, T_{D,I}^h(k), \delta_1) \right) \\
&\overset{(i)}{<} \sum_{h=1}^{H} \sum_{m=1}^{M} \sum_{(D,I) \in \mathcal{T}_k^h} \sum_{T_{D,I}^h(k)=1}^{k} 2 \exp\left[ -\frac{2\beta(k, T_{D,I}^h(k), \delta_1)^2 T_{D,I}^h(k)^2}{T_{D,I}^h(k)(H-h+1)^2} \right] \\
&= \sum_{h=1}^{H} \sum_{m=1}^{M} \sum_{(D,I) \in \mathcal{T}_k^h} \sum_{T_{D,I}^h(k)=1}^{k} 2 \exp\left[ -\frac{2\beta(k, T_{D,I}^h(k), \delta_1)^2 T_{D,I}^h(k)}{(H-h+1)^2} \right] \\
&= \sum_{h=1}^{H} \sum_{m=1}^{M} \sum_{(D,I) \in \mathcal{T}_k^h} \sum_{T_{D,I}^h(k)=1}^{k} 2 \exp\left[ -\ln\left( \frac{2Mk \sum_{h=1}^{H} |\mathcal{T}_k^h|}{\delta_1} \right) \right] \\
&= \sum_{h=1}^{H} \sum_{m=1}^{M} \sum_{(D,I) \in \mathcal{T}_k^h} \sum_{T_{D,I}^h(k)=1}^{k} 2 \cdot \frac{\delta_1}{2Mk \sum_{h=1}^{H} |\mathcal{T}_k^h|} \\
&= \delta_1,
\end{aligned}
$$

where $(i)$ is using Hoeffding's inequality. So there exists $\mathbb{P}(\xi_k) \geq 1 - \delta_1$, the proof is completed.

## E    PROOF OF LEMMA 3

From the traverse function, along the path $P_k^h$, we have

$$
\begin{aligned}
B_{D,I}^h(k) &= \max\left[ L_{D,I}^h(k), \min_{j \in \{2I-1, 2I\}} B_{D+1,j}^h(k) \right] & (22) \\
&\geq \min_{j \in \{2I-1, 2I\}} B_{D+1,j}^h(k) \\
&= B_{D+1,I'}^h(k) \left( (D+1, I') \in P_k^h \right).
\end{aligned}
$$

We make an assumption that at the step $h$ in episode $k$, the attacker launches the attack and chooses a node $(D, I)$ along the path $P_k^h$, then we have

$$
B_{D',I'}^h(k) \geq B_{D,I}^h(k) \geq L_{D,I}^h(k) \left( D' < D, (D', I') \in P_k^h \right). \tag{23}
$$

Because the root node includes $a_h^-(s_h^k)$, so in the path $P_k^h$, except $(D, I)$, there must exist a node $(D_{min}, I_{min})(D_{min} < D)$ containing action $a_h^-(s_h^k)$. So we have

$$
B_{D_{min}, I_{min}}^h(k) \geq B_{D,I}^h(k) \geq L_{D,I}^h(k) \tag{24}
$$

established. Otherwise, from the definition of $L_{D,I}^h(t)$, we can obtain

$$
\begin{aligned}
L_{D_{min}, I_{min}}^h(k) &= \hat{Q}_{D_{min}, I_{min}}^h(s_h^k, a_{D_{min}, I_{min}}) - \beta\left( k, T_{D_{min}, I_{min}}^h(k), \delta_1 \right) - L_s d_s - \nu_1 \rho^{D_{min}} \\
&\overset{(i)}{\leq} \mathbb{E}\left[ \hat{Q}_{D_{min}, I_{min}}^h(s_h^k, a_{D_{min}, I_{min}}) \right] - L_s d_s - \nu_1 \rho^{D_{min}} \\
&\leq Q_h^\dagger(s_h^k, a_{D_{min}, I_{min}}) - \nu_1 \rho^{D_{min}} \\
&\leq Q_h^\dagger(s_h^k, a_h^-(s_h^k)),
\end{aligned}
$$

where $(i)$ is under Lemma 2. We assume that a leaf node $(D_m, I_m)$ contains action $a_h^-(s_h^k)$, there exists

$$
B_{D_m, I_m}^h(k) = L_{D_m, I_m}^h(k) \leq Q_h^\dagger(s_h^k, a_h^-(s_h^k)),
$$

and obviously, all the nodes containing action $a_h^-(s_h^k)$ with $H > H_m$ are descendants of $(D_m, I_m)$. Now by propagating the upper bound of nodes containing $a_h^-(s_h^k)$, i.e., $Q_h^\dagger(s_h^k, a_h^-(s_h^k))$ backward

from $(D_m, I_m)$ to $(D_{min}, I_{min})$ through (22), we can show that $Q_h^\dagger(s_h^k, a_h^-(s_h^k))$ is a valid upper bound of $B_{D_{min}, I_{min}}^h$.

Then from (24), we have, i.e.,

$$
\begin{aligned}
Q_h^\dagger(s_h^k, a_h^-(s_h^k)) &\geq B_{D_{min}, I_{min}}^h(k) \geq B_{D,I}^h(k) \geq L_{D,I}^h(k) \\
&= \hat{Q}_{D,I}^h(s_h^k, a_{D,I}) - \beta(k, T_{D,I}^h(k), \delta_1) - L_s d_s - \nu_1 \rho^D \\
&\overset{(i)}{\geq} \mathbb{E}\left[\hat{Q}_{D,I}^h(s_h^k, a_{D,I})\right] - 2\beta(k, T_{D,I}^h(k), \delta_1) - L_s d_s - \nu_1 \rho^D \\
&\overset{(ii)}{\geq} \mathbb{E}\left[\hat{Q}_{D,I}^h(s_h^k, a_{D,I})\right] - 3\beta(k, T_{D,I}^h(k), \delta_1) - L_s d_s,
\end{aligned}
\tag{25}
$$

$(i)$ is under Lemma 2, and $(ii)$ is because the selected node is always a leaf node, which satisfies $\beta(k, T_{D,I}^h(k), \delta_1) > \nu_1 \rho^D$.

Based on (25), we can obtain

$$
\mathbb{E}\left[\hat{Q}_{D,I}^h(s_h^k, a_{D,I})\right] - Q_h^\dagger(s_h^k, a_h^-(s_h^k)) \leq 3\beta(k, T_{D,I}^h(k), \delta_1) + L_s d_s.
\tag{26}
$$

The proof is completed.

## F   PROOF OF THEOREM 2

Since the attacker will not launch the attack when the agent chooses an action satisfying $a_h \in \mathcal{A}_h^\dagger(s_h)$, we have $\overline{Q}_h^o(s_h^k, a_h^k) = Q_h^o(s_h^k, a_h^k) \leq V_h^o(s_h^k)$. Otherwise, the attacker selects a node $(D, I)$ according to the $\hat{Q}$-values, and replaces $a_h^k$ to the corresponding action $a_{D,I}$, i.e., $\overline{Q}_h^o(s_h^k, a_h^k) = \mathbb{E}[\hat{Q}_{D,I}^h(s_h^k, a_{D,I})]$. Then we have

$$
\begin{aligned}
&\sum_{k=1}^K \sum_{h=1}^H \overline{\Delta}_h^k \\
&= \sum_{k=1}^K \sum_{h=1}^H \left(\overline{V}_h^o(s_h^k) - \overline{Q}_h^o(s_h^k, a_h^k)\right) = \sum_{k=1}^K \sum_{h=1}^H \left(V_h^o(s_h^k) - \overline{Q}_h^o(s_h^k, a_h^k)\right) \\
&= \sum_{(k,h)\notin\alpha} V_h^o(s_h^k) - Q_h^o(s_h^k, a_h^k) + \sum_{(k,h)\in\alpha} V_h^o(s_h^k) - \mathbb{E}[\hat{Q}_{D,I}^h(s_h^k, a_{D,I})] \\
&\geq \sum_{(k,h)\in\alpha} V_h^o(s_h^k) - \mathbb{E}[\hat{Q}_{D,I}^h(s_h^k, a_{D,I})] \\
&\overset{(i)}{\geq} \sum_{(k,h)\in\alpha} V_h^o(s_h^k) - Q_h^\dagger(s_h^k, a_h^-(s_h^k)) - 3\beta(k, T_{D,I}^h(k), \delta_1) - L_s d_s \\
&= \sum_{(k,h)\in\alpha} V_h^o(s_h^k) - V_h^\dagger(s_h^k) + V_h^\dagger(s_h^k) - Q_h^\dagger(s_h^k, a_h^-(s_h^k)) - 3\beta(k, T_{D,I}^h(k), \delta_1) - L_s d_s \\
&\geq \sum_{(k,h)\in\alpha} \Delta_\dagger^o + \Delta_{min}^\dagger - 3\beta(k, T_{D,I}^h(k), \delta_1) - L_s d_s \\
&= |\alpha|(\Delta_\dagger^o + \Delta_{min}^\dagger - L_s d_s) - 3\sum_{(k,h)\in\alpha} \beta(k, T_{D,I}^h(k), \delta_1),
\end{aligned}
\tag{27}
$$

where $(i)$ is under Lemma 3. Next, we analyze $\sum_{(k,h)\in\alpha} \beta(k, T_{D,I}^h(k), \delta_1)$. Define $N_{D,I}^h(S_m) := \{\gamma : \gamma \leq K, s_h^\gamma \in S_m, a_h^\gamma = a_{D,I}\}$ as the total number of episodes that in step $h$, the corresponding state belongs to subinterval $S_m$ and the attacker launches the attack and selects node $(D, I)$.

$$\sum_{(k,h)\in\alpha} \frac{1}{\sqrt{T_{D,I}^h(k)}} = \sum_{h=1}^{H}\sum_{m=1}^{M}\sum_{(D,I)\in\mathcal{T}_K^h}\sum_{N=1}^{N_{D,I}^h(S_m)}\frac{1}{\sqrt{N}}$$

$$\leq \sum_{h=1}^{H}\sum_{m=1}^{M}\sum_{(D,I)\in\mathcal{T}_K^h}\int_1^{N_{D,I}^h(S_m)}\frac{1}{\sqrt{N}}dN$$

$$\leq \sum_{h=1}^{H}\sum_{m=1}^{M}\sum_{(D,I)\in\mathcal{T}_K^h}2\sqrt{N_{D,I}^h(S_m)}$$

$$= M\cdot\sum_{h=1}^{H}|\mathcal{T}_K^h|\cdot\frac{1}{M\cdot\sum\limits_{h=1}^{H}|\mathcal{T}_K^h|}\sum_{h=1}^{H}\sum_{m=1}^{M}\sum_{(D,I)\in\mathcal{T}_K^h}2\sqrt{N_{D,I}^h(S_m)}$$

$$\leq 2M\cdot\sum_{h=1}^{H}|\mathcal{T}_K^h|\sqrt{\frac{\sum_{h=1}^{H}\sum_{m=1}^{M}\sum_{(D,I)\in\mathcal{T}_K^h}N_{D,I}^h(S_m)}{M\cdot\sum_{h=1}^{H}|\mathcal{T}_K^h|}}$$

$$= 2\sqrt{M\cdot\sum_{h=1}^{H}|\mathcal{T}_K^h|\sum_{h=1}^{H}\sum_{m=1}^{M}\sum_{(D,I)\in\mathcal{T}_K^h}N_{D,I}^h(S_m)}$$

$$= 2\sqrt{M\sum_{h=1}^{H}|\mathcal{T}_K^h|\cdot|\alpha|}.$$

Then combine (27), we have

$$\sum_{k=1}^{K}\sum_{h=1}^{H}\overline{\Delta}_h^k \geq |\alpha|(\Delta_\dagger^o + \Delta_{min}^\dagger - L_s d_s) - 3\sum_{(k,h)\in\alpha}\beta(k,T_{D,I}^h(k),\delta_1)$$

$$= |\alpha|(\Delta_\dagger^o + \Delta_{min}^\dagger - L_s d_s) - 3\sum_{(k,h)\in\alpha}\frac{H-h+1}{\sqrt{2T_{D,I}^h(k)}}\sqrt{\ln\left(\frac{2Mk\sum_{h=1}^{H}|\mathcal{T}_k^h|}{\delta_1}\right)}$$

$$\geq |\alpha|(\Delta_\dagger^o + \Delta_{min}^\dagger - L_s d_s) - 3\sum_{(k,h)\in\alpha}\frac{H-h+1}{\sqrt{2T_{D,I}^h(k)}}\sqrt{\ln\left(\frac{2MK\sum_{h=1}^{H}|\mathcal{T}_K^h|}{\delta_1}\right)}$$

$$\geq |\alpha|(\Delta_\dagger^o + \Delta_{min}^\dagger - L_s d_s) - 3(H-h+1)\cdot\sqrt{\ln\left(\frac{2MK\cdot HK}{\delta_1}\right)}\cdot\sqrt{2}\cdot\sqrt{M\sum_{h=1}^{H}|\mathcal{T}_K^h|\cdot|\alpha|}$$

$$= |\alpha|\left(\Delta_\dagger^o + \Delta_{min}^\dagger - L_s d_s\right) - 3\sqrt{|\alpha|}(H-h+1)\cdot\sqrt{\ln\left(\frac{2MK\cdot HK}{\delta_1}\right)}\cdot\sqrt{2M\cdot\sum_{h=1}^{H}|\mathcal{T}_K^h|}.$$
$$\tag{28}$$

Use (17) to bound $\sum_{k=1}^{K}\sum_{h=1}^{H}\overline{\Delta}_h^k$, i.e.,

$$\sum_{k=1}^{K}\sum_{h=1}^{H}\overline{\Delta}_h^k \leq \sum_{k=1}^{K}\left(\overline{V}_1^o(s_1^k) - \overline{V}_1^{\pi^k}(s_1^k)\right) + 2H^2\sqrt{\ln(1/\delta_2)\sum_{k=1}^{K}\left(\overline{V}_1^o(s_1^k) - \overline{V}_1^{\pi^k}(s_1^k)\right)}$$

$$\overset{(i)}{\leq} \sum_{k=1}^{K}\left(\overline{V}_1^{\pi^{*,k},k}(s_1^k) - \overline{V}_1^{\pi^k,k}(s_1^k)\right) + 2H^2\sqrt{\ln(1/\delta_2)\sum_{k=1}^{K}(\overline{V}_1^{\pi^{*,k},k}(s_1^k) - \overline{V}_1^{\pi^k,k}(s_1^k))}$$

$$= \text{D-Regret}(K) + 2H^2\sqrt{\log(1/\delta_2)\,\text{D-Regret}(K)},$$
$$\tag{29}$$

where $\pi^{*,k} = \sup_\pi V_1^{\pi,k}(s_1^k)$ is the optimal policy of episode $k$. The reason for $(i)$ is that because of the existence of the attacker, the environment is non-stationary in the observation of the agent. Combine (28) and (29), we can obtain

$$\text{D-Regret}(K) + 2H^2\sqrt{\ln(1/\delta_2)\,\text{D-Regret}(K)} \geq$$

$$|\alpha|\,(\Delta_\dagger^o + \Delta_{\min}^\dagger - L_s d_s) - 3\sqrt{|\alpha|}(H - h + 1)\cdot\sqrt{\ln\left(\frac{2MK\cdot HK}{\delta_1}\right)}\cdot\sqrt{2M\cdot\sum_{h=1}^{H}|\mathcal{T}_K^h|}.$$
$$(30)$$

Finally, we can obtain

$$|\alpha| \leq \frac{\text{D-Regret}(K) + 2H^2\sqrt{\ln(1/\delta_2)\cdot\text{D-Regret}(K)}}{\Delta_\dagger^o + \Delta_{min}^\dagger - L_s d_s} + \frac{18MH^2\ln(\frac{2MHK^2}{\delta_1})\sum_{h=1}^{H}|\mathcal{T}_K^h|}{(\Delta_\dagger^o + \Delta_{min}^\dagger - L_s d_s)^2},$$
$$(31)$$

with $|\tau| \leq |\alpha|$, the proof is completed.

### F.1 THE NODE NUMBER OF THE COVER TREE

In this subsection, we bound the node number of the cover tree, i.e., $|\mathcal{T}_K^h|$ for $\forall h \in [1, H]$.

A node $(D, I)$ will be expanded when it satisfies the condition

$$\nu_1\rho^D \geq \frac{H - h + 1}{\sqrt{2T_{D,I}^h(k)}}\cdot\sqrt{\ln\left(\frac{2Mk\cdot\sum_{h=1}^{H}|\mathcal{T}_k^h|}{\delta_1}\right)}.$$

We make some transformations, i.e.,

$$T_{D,I}^h(k) \geq \frac{(H - h + 1)^2}{2\nu_1^2\rho^{2D}}\cdot\ln\left(\frac{2Mk\cdot\sum_{h=1}^{H}|\mathcal{T}_k^h|}{\delta_1}\right)$$

$$> \frac{(H - h + 1)^2}{2\nu_1^2\rho^{2D}}\cdot\ln\left(\frac{2M\cdot 3H}{\delta_1}\right) = \frac{(H - h + 1)^2}{2\nu_1^2\rho^{2D}}\cdot\ln\left(\frac{6MH}{\delta_1}\right).$$

From the inequality above, we can see that as the depth $D$ increases, the $T_{D,I}^h(k)$-value which can make the inequality above established will also increase, so it's obvious that when the tree $\mathcal{T}_k^h$ is a complete binary tree, its total number of nodes is the most. Based on this, we assume that the depth is $D_m$, i.e., the nodes at depth $1, 2, ..., D_m - 1$ have been expanded, then we have

$$K \geq \sum_{D=1}^{D_m-1} 2^D\cdot\frac{(H - h + 1)^2}{2\nu_1^2\rho^{2D}}\cdot\ln\left(\frac{6MH}{\delta_1}\right)$$

$$= \sum_{D=1}^{D_m-1}\left(2\rho^{-2}\right)^D\cdot\frac{(H - h + 1)^2}{2\nu_1^2}\cdot\ln\left(\frac{6MH}{\delta_1}\right)$$

$$= \frac{(H - h + 1)^2}{\nu_1^2}\cdot\ln\left(\frac{6MH}{\delta_1}\right)\cdot\frac{\left(2\rho^{-2}\right)^{D_m-1} - 1}{2 - \rho^2}.$$

Then we can get the upper bound of $D_m$, i.e.,

$$D_m \leq \log_{2\rho^{-2}}\left[\frac{K\cdot\nu_1^2\cdot\left(2 - \rho^2\right)}{(H - h + 1)^2\cdot\ln\left(6MH/\delta_1\right)} + 1\right] + 1.$$
$$(32)$$

Through $2^{D_m+1} - 1$, we get the upper bound of the node number of tree $\mathcal{T}_K^h$, i.e.,

$$|\mathcal{T}_K^h| \leq 4\left[\frac{K\cdot\nu_1^2\cdot\left(2 - \rho^2\right)}{(H - h + 1)^2\cdot\ln\left(6MH/\delta_1\right)} + 1\right]^{\log_{2\rho^{-2}}2} = O\left(K^{\log_{2\rho^{-2}}2}\right).$$
$$(33)$$

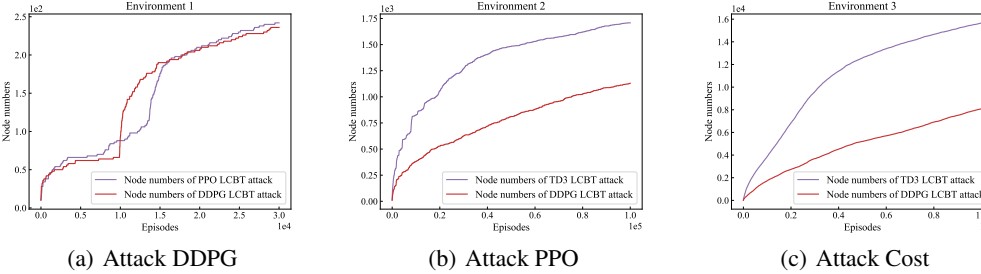

Figure 6: *The total number of nodes of the cover trees.*

Note that because $0 < \rho < 1$, we always have $\log_{2\rho^{-2}} 2 < 1$. At the same time, based on the characteristics of the LCBT algorithm itself, it is highly unlikely to have a complete binary tree due to the algorithm's preference for expanding nodes containing action $a_h^-(s_h^k)$, with only one such node in the same depth.

In Fig. 6, the graph shows the relationship between the number of episodes and the total number of nodes of all the cover trees, i.e., $\sum_{h=1}^{H} |\mathcal{T}_k^h|$. The x-axis indicates the number of episodes during the training phase, while the y-axis represents the total number of nodes of all the cover trees after each corresponding episode. It is evident that compared to the increase in the number of episodes, the increase in the number of nodes is slow, and the final number of nodes is far smaller than the number of total episodes. In addition, The number of nodes within all the cover trees is not primarily impacted by the algorithm chosen by the agent, but instead depends more upon the environment in which the agent is placed, as well as the hyperparameters applied to the LCBT attack algorithm.

## G  ADDITIONAL RESULTS

We measured the percentage of target actions executed by the attacked policy in identical states as a metric for quantifying the similarity between the attacked and target policies. The results are shown in Table 2 and Table 3.

In these two tables, the similarity calculation formula we adopt is as follows:

$$sim = \frac{\sum(\mathbb{I}\{l_a(a_1, a_2) < r_a\})}{steps}$$

The $steps$ refers to the total number of steps during the similarity testing period. In this case, $steps = 1.0 * 10^5$. $a_1$ represents the action output of the target policy, while $a_2$ represents the action output of the attacked policy. $l_a(\cdot, \cdot)$ denotes the distance between the two actions, where the Euclidean distance is used. For the indicator function $\mathbb{I}\{\xi\}$, if event $\xi$ is established $\mathbb{I}\{\xi\} = 1$, otherwise $\mathbb{I}\{\xi\} = 0$.

From the tables, we can observe that the attacked policy shows a high degree of similarity with the target policy. Similarly, as the complexity of the environment increases, the similarity tends to decrease accordingly. In the same environment, different algorithms and corresponding $D - Regret$ can also have an impact on the results.

Additionally, we have introduced a new experiment in environment 3. The corresponding experimental outcomes are presented in Fig. 7. Remarkably, under the conditions of a 5-dimensional space, the LCBT algorithm demonstrates commendable performance. Hence, in practical terms, factors influencing the efficacy of the LCBT algorithm encompass not only its intrinsic parameters such as $\nu_1$, $\rho$, $d_s$, but also encompass the environment and the algorithms employed by the agent. Similarly, in Table 4, we have provided the results concerning the similarity between the attack strategy and the target strategy. We recorded the time taken to execute the attack algorithm and the total time during the experimental process, calculated the ratio between the two, and obtained the results as depicted in Fig. 8.

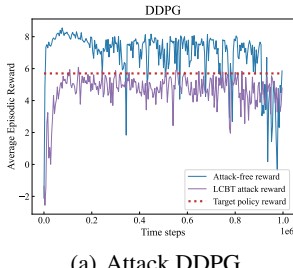
(a) Attack DDPG

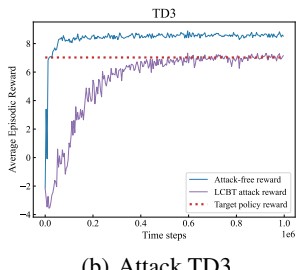
(b) Attack TD3

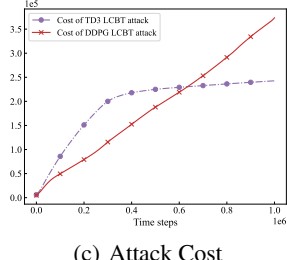
(c) Attack Cost

Figure 7: *Reward and cost results of Environment 3. In this experiment, we set $r_a = 0.497$, $M = 59049$, $H = 10$, and $T = 10^6$. $\rho$ is set to be $1/\sqrt{5}$.*

Table 2: *The similarity between the original target policy and attacked policy in environment 1. We have selected attacked policies trained for $3 * 10^5$ steps and $4 * 10^5$ steps under either Oracle attack or LCBT attack. Then, in the testing phase, we calculated the percentage of target actions executed by the attacked strategy within $1 * 10^5$ steps. In this experiment, we set the similarity radius $r_a = 0.0625$, which is consistent with the training phase mentioned in the paper.*

| | Training steps | DDPG | | PPO | |
|---|---|---|---|---|---|
| | | Oracle attack | LCBT attack | Oracle attack | LCBT attack |
| Environment 1 | 3e5 | 99.339% | 97.884% | 99.930% | 99.920% |
| | 4e5 | 99.828% | 99.787% | 100.00% | 100.00% |

Table 3: *The similarity between the original target policy and attacked policy in environment 2. Using a similar method as in Table 2, we separately select the attacked policies for $1.0 * 10^6$ steps and $1.8 * 10^6$ steps and match their similarity with the original target policy. In this experiment, we set the similarity radius $r_a = 0.31$, which is consistent with the training phase mentioned in the paper.*

| | Training steps | DDPG | | TD3 | |
|---|---|---|---|---|---|
| | | Oracle attack | LCBT attack | Oracle attack | LCBT attack |
| Environment 2 | 1e6 | 85.502% | 76.771% | 99.864% | 98.564% |
| | 1.8e6 | 90.471% | 82.421% | 99.914% | 99.076% |

Table 4: *The similarity between the original target policy and attacked policy in environment 3. Using a similar method as in Table 2, we select the attacked policies for $8 * 10^5$ steps and match their similarity with the original target policy. In this experiment, we set the similarity radius $r_a = 0.497$, which is consistent with the training phase mentioned in the paper.*

| | Training steps | DDPG LCBT attack | TD3 LCBT attack |
|---|---|---|---|
| Environment 3 | 8e5 | 78.488% | 97.812% |

Table 5: *A comparison between the two attack algorithms. $\mathcal{R}(T)$ is the upper bound of D-Regret$(K)$ of the agent's RL algorithm, where $T = KH$.*

| Attack Algorithm | Scenario | Cost Bound |
|---|---|---|
| *LCB-H* Liu & Lai (2021) | Discrete state and action spaces | $\dfrac{H\left(\mathcal{R}(T)+2H^2\sqrt{\log(1/p)\mathcal{R}(T)}\right)}{\Delta_{min}} + \dfrac{307SAH^4\log(2SAT/p)}{\Delta_{min}^2}$ |
| *LCBT (Ours)* | Continuous state and action spaces | $\dfrac{\mathcal{R}(T)+2H^2\sqrt{\ln(1/\delta_2)\cdot\mathcal{R}(T)}}{\Delta_\dagger^o+\Delta_{min}^\dagger-L_s d_s} + \dfrac{18MH^2\ln(\frac{2MHK^2}{\delta_1})\sum_{h=1}^H |\mathcal{T}_K^h|}{(\Delta_\dagger^o+\Delta_{min}^\dagger-L_s d_s)^2}$ |

# H DISCUSSION

In this section, we will compare our work with that in Liu & Lai (2021), which focuses on the action-manipulation attack in discrete state and action spaces. Table 5 represents the cost bounds

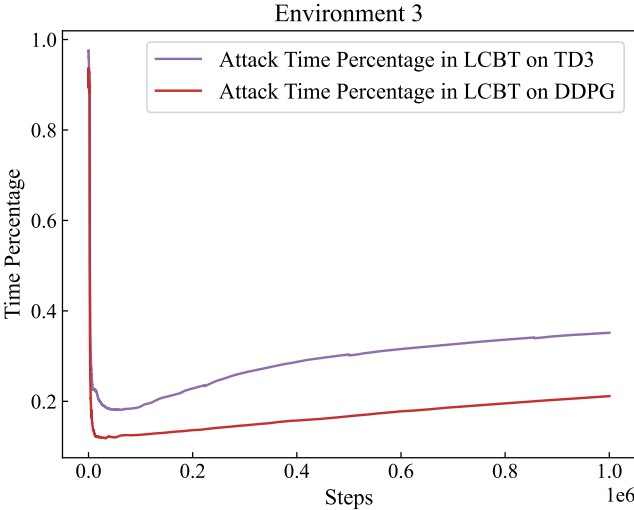

Figure 8: *In the initial phase, the reinforcement learning algorithm undergoes a completely random exploration period, during which the algorithm consumes minimal time. As a result, the time proportion for LCBT attack is notably high. Subsequently, as the neural network within the reinforcement learning process intervenes, this proportion rapidly decreases. In the subsequent phases, due to the increasing number of nodes in the LCBT algorithm's tree, the proportion of time consumed by running the LCBT algorithm gradually escalates.*

for LCB-H attack Liu & Lai (2021) and our LCBT attack. In the LCB-H attack, the denominator is $\Delta_{min} = \min_{h \in [H], s \in S} \left( V_h^{\dagger}(s) - \min_{a \in A} Q_h^{\dagger}(s, a) \right)$, whose definition is the same as $\Delta_{min}^{\dagger}$ in our work. In our denominator, $\Delta_{\dagger}^{o}$ represents the gap between the policies $\pi^o$ and $\pi^{\dagger}$, and $L_s d_s$ represents the uncertainty generated by the state subinterval. In addition, there both exists $\mathcal{R}(T) + 2H^2\sqrt{\log(1/p)\mathcal{R}(T)}$ in the two cost bounds because the effectiveness of the algorithm used by the agent has an important impact on the cost. In other words, the agent with a more efficient algorithm can learn the target policies faster when under attack. In LCB-H, The reason for multiplying with $H$ is that when the attacker launches an attack, the original action has only a $1/H$ probability of being replaced with the worst action, otherwise, it will be replaced with the target action. The number of states $S$ in LCB-H is replaced by the number of subintervals $M$ in LCBT. And the number of actions $A$ is replaced by the total node number $\sum_{h=1}^{H} |\mathcal{T}_K^h|$ while we use the cover trees to discretize the action space.

