# OpenReview forum: "Tree-based Action-Manipulation Attack Against Continuous Reinforcement Learning with Provably Efficient Support"
_ICLR.cc/2024/Conference — ICLR 2024 Conference Withdrawn Submission_

### Official Review · Reviewer_BCGQ · 2023-10-28

**Soundness:** 3 good
**Presentation:** 2 fair
**Contribution:** 2 fair
**Rating:** 6
**Confidence:** 2

**Summary:**

The paper presents a new threat model for targeted action space attacks on RL algorithms in continuous domains. The authors first developed the threat model in a white box setting, then proposed an algorithm for the black-box setting which performs similar to the white box setting. This method leverages the idea of recursively partitioning the continuous state and action spaces into a binary tree, where each node represents a continuous subset of actions and selecting the nodes from the tree during the attack. The authors also provided theoretical results which shows that the attack costs are bounded under specific conditions in the white box setting. Finally, numerical experiments were provided to three popular RL algorithms in two simple environments.

**Strengths:**

1. The idea of partitioning a continuous space into a binary tree is an interesting idea and original in the space of RL to the best of my knowledge.
2. The algorithm for LCBT is theoretically motivated with a bound on the cost of the attack
3. The numerical experiments seems promising and results supports the theory

**Weaknesses:**

1. Insufficient experimental results in terms of comparison with other action space manipulation threat models.
2. Insufficient experimental results in terms of comparison in more complex, realistic environments.
3. Multiple generic statements which might not necessarily be true.

Please refer to the next section for detail comments.

**Questions:**

I would like to preface my comments by stating that I think the algorithm proposed by this paper is an interesting idea, however, my concerns are as follows:

1. As stated above, one of my main concerns is the scalability of the algorithm as the numerical experiments were only shown for two extremely simple environments. Could the authors discuss the computational runtime for such the algorithm? It seemed to me that constructing the binary tree for environments which higher dimensional state spaces/action spaces would be prohibitively expensive. Furthermore, computation aside, the results would be more convincing and much stronger if the authors could show similar results on a much more conventional benchmark, like the MuJoCo environments.

2. The results shown are also lacking comparison with existing threat models, even in terms of white-box attacks, such as the work by Sun et al. 2020.

3. There's also sentences in the introduction which I believe the authors should further clarify as they seem like blanket statements to me which might not necessarily be true. For example, "With the increasing complexity of scenarios, in many cases, reinforcement learning algorithms for discrete state-action environments are no longer applicable". This seems like a really harsh statement as there is a lot of real world problems where discrete actions are still applicable. Another statement is  "It is evident that compared to manipulating observations, rewards, or the environment, action-manipulation is not as direct and efficient". Could the authors elaborate why action space manipulation is more challenging than reward/state manipulation as this seem to be the core motivation for action space attacks over other forms of attack?

**Details Of Ethics Concerns:**

The threat models described in this paper could potentially be applied to real world systems.

---

> ### Author Response · Authors · 2023-11-20
>
> | Method          | Attacker's goal                                              | The support for continuous action space | The demand for knowledge of the MDP | The demand for knowledge of the RL algorithm used by the agent | The adversary's requisition for neural networks or gradient computations during attack | Attack cost |
> | --------------- | ------------------------------------------------------------ | --------------------------------------- | ----------------------------------- | ------------------------------------------------------------ | ------------------------------------------------------------ | ----------- |
> | LCB-H[1]        | Make the agent learn a target policy                         | No                                      | No                                  | No                                                           | No                                                           | Given       |
> | LAS[2]          | Minimize the rewards                                         | Yes                                     | Yes                                 | Yes                                                          | Yes                                                          | Not Given   |
> | VA2C-P[3]       | Make the agent learn a target policy or Minimize the rewards | Yes                                     | No                                  | Yes                                                          | Yes                                                          | Not Given   |
> | LCBT (our work) | Make the agent learn a target policy                         | Yes                                     | No                                  | No                                                           | No                                                           | Given       |
>
> [1] Provably Efficient Black-Box Action Poisoning Attacks Against Reinforcement Learning. NIPS 2021.
>
> [2] Spatiotemporally constrained action space attacks on deep reinforcement learning agents. AAAI 2020.
>
> [3] Vulnerability-aware poisoning mechanism for online rl with unknown dynamics. ICLR 2021.

---

> ### Author Response · Authors · 2023-11-21
>
> Thanks for you feedback. I will response to the questions you mentioned.
>
> ## Question 1: As stated above, one of my main concerns is the scalability of the algorithm as the numerical experiments were only shown for two extremely simple environments. Could the authors discuss the computational runtime for such the algorithm? It seemed to me that constructing the binary tree for environments which higher dimensional state spaces/action spaces would be prohibitively expensive. Furthermore, computation aside, the results would be more convincing and much stronger if the authors could show similar results on a much more conventional benchmark, like the MuJoCo environments.
>
> ### 1.computational runtime
>
> The computational time complexity of the LCBT algorithm primarily emanates from two aspects:
>
> (1) After each round, the attacker needs to update the corresponding $L$ and $B$ values for each node of every tree, and the complexity of this step is related to the number of nodes in the tree. Specifically, assuming that in the k-th episode, after completion, the attacker updates each node of H trees, according to the proof in Appendix F.1, the upper bound on the number of nodes in a tree is $O(k^E)$, where $E=\log_{2\rho^{-2}}2<1$. Therefore, for the total K episodes, the time complexity of this part is $O(K\cdot H\cdot K^E)$.
>
> (2) When the attacker launches an attack, it is necessary to traverse from the root node to the leaf node (the WorTraverse function), and the complexity of this step is related to the depth of the tree. By employing a proof method akin to that in Appendix F.1, we can ascertain an upper bound for the depth of the tree as $O(\log_{\rho^{-2}}k)$ in episode $k$. Making an extreme assumption that the attacker launches an attack at every step, the time complexity for this part over the total K episodes is $O(K\cdot H \cdot \log_{\rho^{-2}}K)$.
>
> In summary, the overall computational time complexity is given by: $O(H\cdot K^{1+E}+H\cdot K \cdot \log_{\rho^{-2}}K)$.
>
> ### 2.More conventional benchmark
>
> We conducted an additional experiment in a five-dimensional space, the environmental setup of which is detailed in Environment 3 within Appendix A. The corresponding experimental outcomes are presented in Figure 7 of Appendix G.
>
> ## Question 2: The results shown are also lacking comparison with existing threat models, even in terms of white-box attacks, such as the work by Sun et al. 2020.
>
> In the above table, we conducted a comparative analysis of existing literature concerning attack methodologies centered on action manipulation. Notably, our comparison included the metric "The adversary's requisition for neural networks or gradient computations during attack," setting it as one of the distinctions between our approach and other studies. While existing works typically implement attacks from the neural network perspective in continuous action spaces, we consider the neural network as a black box, constructing attacks solely from the theoretical framework of reinforcement learning itself. This distinction forms a key aspect of our current work in contrast to others in the field. Hence, our work did not directly compare with other studies, such as [3], in the experimental setup.
>
> ## Question 3: Could the authors elaborate why action space manipulation is more challenging than reward/state manipulation as this seem to be the core motivation for action space attacks over other forms of attack?
>
> In the context of action-manipulation attacks, the fundamental factor influencing the intelligent agent lies in the attacker's manipulation of actions, thereby impacting rewards and consequently leading to the potential misguidance of the intelligent agent. Therefore, the action-manipulation attack method can be viewed as indirectly influencing the training of the intelligent agent by altering rewards. In contrast to the direct manipulation of observations, rewards, or the environment, action-manipulation represents a less direct approach. Additionally, a salient aspect of action-manipulation attacks is the constrained operational domain available to attackers, limited to the innate action space of the intelligent agent. In contrast, adopting reward manipulation allows attackers to modify rewards to values lower than those associated with the worst actions, presenting a more conspicuous avenue for manipulation.
>
> Additionally, in black-box conditions, the method of manipulating rewards often only requires providing a minimal reward value during the attack, whereas finding an approximate worst-case action in a specific state is not a straightforward task.

---

> > ### Comment · Reviewer_BCGQ · 2023-11-21
> >
> > I would like to thank the authors for their detailed response.
> >
> > 1. In addition to the theoretical bound, could the authors provide an actual wall-clock runtime on the experiments as well together with the associated hardware specifications used to mount the attacks? I am still not entirely convinced that this is a scalable algorithm in practice and I am more than happy to change my decision if the authors could address this in more detail. Also, for completeness, the authors should also provide a table similar to Table 2 and 3 in the appendix for the newly added experiment.
> >
> > 2. I accept the authors' rebuttal to this point and have no further queries.
> >
> > 3. I still do not follow how is action space manipulation more challenging than other forms of manipulation. State manipulation can also be viewed as an indirectly influencing the training of the intelligent agent by altering the rewards and is also not as direct as reward manipulation. The additional points that the authors provide in support of action manipulation similarly applies to state manipulation as well, which does not make action manipulation more challenging than state manipulation.

---

> ### Author Response · Authors · 2023-11-22
>
> ## Comment 1.
>
> ### 1. runtime
>
> We reran the experiments for the LCBT attack corresponding to Environment 3, recording the time taken to execute the attack algorithm and the total time. The results are presented in the table below, wherein the **"percentage"** represents the ratio of the time taken to execute the attack algorithm up to that point compared to the total time.
>
> |          | **Steps** | **Attack Time(s)** | **Total Time(s)** | **Percentage** |
> | -------- | --------- | ------------------ | ----------------- | -------------- |
> | **DDPG** | 5e5       | 659                | 3948              | 16.71%         |
> |          | 1e6       | 1763               | 8338              | 21.14%         |
> | **TD3**  | 5e5       | 1329               | 4379              | 30.35%         |
> |          | 1e6       | 3375               | 9599              | 35.16%         |
>
> Additionally, we included a figure (Figure 8) in Appendix G displaying the variation of the **"percentage"** metric with respect to the number of steps.
>
> ### 2. scalability
>
> Here, we shall begin by conducting a theoretical analysis of the circumstances surrounding the method proposed by us in high-dimensional settings. One obvious point is that our method requires partitioning of the action and state spaces. With higher dimensions in the action space, even if we use binary tree partitioning, the sufficiency of partitioning will still be lower than in lower dimensions with the same number of partitions. This means that the LCBT algorithm converges slower towards the worst action, resulting in a higher level of non-stationarity in the environment and a larger **D-Regret**. According to Theorem 2, this also leads to higher attack cost.
>
> Additionally, in the new experiment, the LCBT algorithm demonstrates commendable performance. Hence, in practical terms, factors influencing the efficacy of the LCBT algorithm encompass not only its intrinsic parameters such as $\nu_1$, $\rho$, $d_s$ (For instance, the settings of $\nu_1$ and $\rho$ can influence the speed of tree node expansion), but also encompass the environment and the algorithms employed by the agent.
>
> ### 3. completeness
>
> We have included the following table in Appendix G.
>
> The similarity between the original target policy and attacked policy in environment 3. Using a similar method as in Table 2, we select the attacked policies for $8\cdot 10^{5}$ steps and match their similarity with the original target policy. In this experiment, we set the similarity radius $r_a=0.497$, which is consistent with the training phase mentioned in the paper. Then, in the testing phase, we calculated the percentage of target actions executed by the attacked strategy within $1\cdot 10^5$  steps.
>
> |                   | **Training steps** | **DDPG LCBT attack** | **TD3 LCBT attack** |
> | ----------------- | ------------------ | -------------------- | ------------------- |
> | **Environment 3** | 8e5                | 78.488%              | 97.812%             |
>
> ## Comment 3.
>
> For this matter, tampering with actions inevitably affects subsequent states; however, the extent of this influence might be less significant than directly altering the state itself. Additionally, considering action manipulation solely as an existing means of attack, it warrants further research and exploration. Lastly, concerning the issue you raised, we will also contemplate refining our presentation in the paper.

---

> > ### Comment · Reviewer_BCGQ · 2023-11-22
> >
> > Thank you for the clarification. Since the authors have address my queries, I'm raising my score from 5 to 6. I do hope the authors will consider refining their presentation of the paper and include the additional details.

---

### Official Review · Reviewer_5Edg · 2023-10-30

**Soundness:** 3 good
**Presentation:** 2 fair
**Contribution:** 3 good
**Rating:** 6
**Confidence:** 4

**Summary:**

This paper investigates adversarial attacks in continuous reinforcement learning environments. The primary focus is on action manipulation attacks, where attackers intercept and modify the actions taken by the intelligent agent before they reach the environment. The paper makes the following key contributions:

1. The paper introduces a threat model for action manipulation attacks in continuous state and action spaces, defining the attacker's goals, knowledge, and capabilities with the help of a target policy. It also introduces the concept of a "target action space" to adapt to continuous environments.

2. The research covers both white-box and black-box scenarios. In the white-box scenario, the attacker has extensive knowledge of the underlying processes, allowing for intuitive attack methods. In the black-box scenario, a novel (as far as the reviewer concerned) attack method called "Lower Confidence Bound Tree" (LCBT) is introduced to approximate the effectiveness of white-box attacks.

3. In the white-box scenario, the paper proposes the "oracle attack" method, which can compel agents using sub-linear-regret reinforcement learning algorithms to select actions that follow target policies with sublinear attack costs.

4. The proposed attack methods are applied to popular RL algorithms, including DDPG, PPO, and TD3, and their effectiveness is demonstrated through experiments.

In summary, this paper addresses the crucial issue of security in continuous reinforcement learning environments, particularly focusing on action manipulation attacks. It provides a comprehensive understanding of these attacks, introduces new attack methods, and validates their effectiveness through experiments.

**Strengths:**

This paper possesses several notable strengths:

_Originality_:

1. The paper addresses a relatively uncharted issue within the field of reinforcement learning, concentrating on adversarial attacks within continuous action spaces. While adversarial attacks in RL have been explored to some extent, the particular focus on action manipulation attacks in continuous environments represents a good contribution.

2. The introduction of the "oracle attack" for white-box scenarios and the "Lower Confidence Bound Tree" (LCBT) for black-box scenarios presents new methods for action manipulation attacks. These approaches offer new viewpoints and solutions to the problem, enhancing the paper's originality.

_Quality_:

3. The paper establishes a solid theoretical foundation for its attack methods, discussing the conditions under which these attacks are effective. The analysis of sub-linear-regret reinforcement learning algorithms and the associated cost bounds adds to the quality of the research.

**Weaknesses:**

While the paper showcases numerous strengths, it also reveals specific limitations that, if addressed, could enhance its quality and influence:

1. The paper's efficacy could be improved through a more thorough examination of prior research in the realm of adversarial attacks on reinforcement learning. While the paper briefly references previous work, a more detailed comparison with existing attack techniques on continuous action spaces and their inherent limitations would serve to underscore the novelty and merits of the proposed methods.

2. Although the paper introduces inventive attack methodologies, the potential complexity of these methods could pose practical challenges for their implementation. Specifically, it would be beneficial to provide an in-depth analysis of the computational complexity associated with the oracle and tree-based attacks.

**Questions:**

1. Where can we get the target policy? How to specify a target policy in a complex task?

2. Could the authors provide more experiments on complex tasks?

---

> ### Author Response · Authors · 2023-11-21
>
> | Method          | Attacker's goal                                              | The support for continuous action space | The demand for knowledge of the MDP | The demand for knowledge of the RL algorithm used by the agent | The adversary's requisition for neural networks or gradient computations during attack | Attack cost |
> | --------------- | ------------------------------------------------------------ | --------------------------------------- | ----------------------------------- | ------------------------------------------------------------ | ------------------------------------------------------------ | ----------- |
> | LCB-H[1]        | Make the agent learn a target policy                         | No                                      | No                                  | No                                                           | No                                                           | Given       |
> | LAS[2]          | Minimize the rewards                                         | Yes                                     | Yes                                 | Yes                                                          | Yes                                                          | Not Given   |
> | VA2C-P[3]       | Make the agent learn a target policy or Minimize the rewards | Yes                                     | No                                  | Yes                                                          | Yes                                                          | Not Given   |
> | LCBT (our work) | Make the agent learn a target policy                         | Yes                                     | No                                  | No                                                           | No                                                           | Given       |
>
> [1] Provably Efficient Black-Box Action Poisoning Attacks Against Reinforcement Learning. NIPS 2021.
>
> [2] Spatiotemporally constrained action space attacks on deep reinforcement learning agents. AAAI 2020.
>
> [3] Vulnerability-aware poisoning mechanism for online rl with unknown dynamics. ICLR 2021.

---

> ### Author Response · Authors · 2023-11-21
>
> Thanks for your feedback. I will response to the weaknesses and questions you mentioned.
>
> ## Weakness 1: The paper's efficacy could be improved through a more thorough examination of prior research in the realm of adversarial attacks on reinforcement learning. While the paper briefly references previous work, a more detailed comparison with existing attack techniques on continuous action spaces and their inherent limitations would serve to underscore the novelty and merits of the proposed methods.
>
> In the above table, we conducted a comparative analysis of existing literature concerning attack methodologies centered on action manipulation. Notably, our comparison included the metric "The adversary's requisition for neural networks or gradient computations during attack," setting it as one of the distinctions between our approach and other studies. While existing works typically implement attacks from the neural network perspective in continuous action spaces, we consider the neural network as a black box, constructing attacks solely from the theoretical framework of reinforcement learning itself. This distinction forms a key aspect of our current work in contrast to others in the field.
>
> Additionally, regarding the algorithm LCB-H[1] in discrete environments, we conducted a more detailed comparison in Appendix H. This examination primarily focuses on exploring the fundamental reasons for the discrepancies between the two in terms of attack cost.
>
> ## Weakness 2: Although the paper introduces inventive attack methodologies, the potential complexity of these methods could pose practical challenges for their implementation. Specifically, it would be beneficial to provide an in-depth analysis of the computational complexity associated with the oracle and tree-based attacks.
>
> Regarding the oracle attack, the adversary is already cognizant of the worst action, thereby enabling a direct manipulation replacement during the attack. At any given step of launching the attack, the time complexity is $O(1)$, and for the overall $K$ episodes, the computational time complexity amounts to $O(|\tau|)$.
>
> The computational time complexity of the LCBT algorithm primarily emanates from two aspects:
>
> (1) After each round, the attacker needs to update the corresponding $L$ and $B$ values for each node of every tree, and the complexity of this step is related to the number of nodes in the tree. Specifically, assuming that in the k-th episode, after completion, the attacker updates each node of H trees, according to the proof in Appendix F.1, the upper bound on the number of nodes in a tree is $O(k^E)$, where $E=\log_{2\rho^{-2}}2<1$. Therefore, for the total K episodes, the time complexity of this part is $O(K\cdot H\cdot K^E)$.
>
> (2) When the attacker launches an attack, it is necessary to traverse from the root node to the leaf node (the WorTraverse function), and the complexity of this step is related to the depth of the tree. By employing a proof method akin to that in Appendix F.1, we can ascertain an upper bound for the depth of the tree as $O(\log_{\rho^{-2}}k)$ in episode $k$. Making an extreme assumption that the attacker launches an attack at every step, the time complexity for this part over the total K episodes is $O(K\cdot H \cdot \log_{\rho^{-2}}K)$.
>
> In summary, the overall computational time complexity is given by: $O(H\cdot K^{1+E}+H\cdot K \cdot \log_{\rho^{-2}}K)$.
>
> ## Question 1: Where can we get the target policy? How to specify a target policy in a complex task?
>
> The essence of a policy lies in mapping states to actions. Therefore, when configuring the target policy, attackers can partition the states into sub-intervals according to their objectives. Each sub-interval corresponds to a target action, or attackers can choose to set target actions only for certain states while leaving the rest unaffected by attacks. In our experiments, we prefer to use a stricter target policy where each state has a corresponding target action. This allows us to evaluate the effectiveness of our algorithm. To achieve this, we introduce additional constraints in the original environment. For instance, in environment 2, we provide a negative reward when the distance between the car and the target point < 1. By training a supplementary model with these extra constraints, we obtain an additional model that represents the attacker's target policy.
>
> ## Question 2: Could the authors provide more experiments on complex tasks?
>
> We conducted an additional experiment in a five-dimensional space, the environmental setup of which is detailed in Environment 3 within Appendix A. The corresponding experimental outcomes are presented in Figure 7 of Appendix G.

---

### Official Review · Reviewer_hNCZ · 2023-10-31

**Soundness:** 2 fair
**Presentation:** 2 fair
**Contribution:** 2 fair
**Rating:** 6
**Confidence:** 3

**Summary:**

This paper presents a new attack algorithm for reinforcement learning agents in
the action-manipulation setting. First, a white-box attack is developed which
assumes access to the underlying MDP, with which an optimal attack action can be
computed. Theoretical bounds show that with high probability, the attacker can
force the agent to approximately learn a target policy with a number of
interventions roughly proportional to the regret of the policy. A black-box
approximation to the attack is then developed which relaxes the assumption of
full knowledge of the MDP. The black box attack uses an action-space
partitioning algorithm to approximate the worst-case action for a given state.
Theoretical bounds for the black box attack show similar results to the
white-box version. Finally, empirical experiments show that the proposed attack
is successful at forcing several common RL algorithms to learn the target
policies.

**Strengths:**

The theoretical results are interesting and (to the best of my knowledge) novel.
In particular, the cost bound for the practical algorithm (Theorem 2) is an
interesting result with important implications for robust RL.

Extending RL attacks to continuous state and action spaces is an important
problem, given how many realistic scenarios are continuous. The discretization
approach to this problem is interesting and the online abstraction refinement
approach is a reasonable way to trade off cost and precision.

**Weaknesses:**

I have some concerns about the experimental evaluation:

- The environments considered are extremely low-dimensional, even by the
  standards of formal methods research.
- There are no comparisons with existing RL attack methods, so it's hard to
  gauge the empirical effectiveness of this approach.
- There are only two environments.

There are a few places where the writing was not very clear to me, most notably
in Section 4.2. I think this section could do with some more intuition or an
example.

**Questions:**

Does the analysis hold for stochastic policies? I was a bit unclear on this
since the definitions of $Q$ and $V$ both use the notation introduced for
deterministic policies. Moreover it seems to me that some parts of the proofs
rely on the assumption that all policies are deterministic (including the
exploration policies of the RL algorithm), but most RL algorithms do not work by
evaluating deterministic policies.

---

> ### Author Response · Authors · 2023-11-20
>
> Thanks for your feedback. I will response to the weaknesses and questions you mentioned.
>
> ## Weaknesses
>
> ### There are no comparisons with existing RL attack methods, so it's hard to gauge the empirical effectiveness of this approach.
>
> | Method          | Attacker's goal                                              | The support for continuous action space | The demand for knowledge of the MDP | The demand for knowledge of the RL algorithm used by the agent | The adversary's requisition for neural networks or gradient computations during attack | Attack cost |
> | --------------- | ------------------------------------------------------------ | --------------------------------------- | ----------------------------------- | ------------------------------------------------------------ | ------------------------------------------------------------ | ----------- |
> | LCB-H[1]        | Make the agent learn a target policy                         | No                                      | No                                  | No                                                           | No                                                           | Given       |
> | LAS[2]          | Minimize the rewards                                         | Yes                                     | Yes                                 | Yes                                                          | Yes                                                          | Not Given   |
> | VA2C-P[3]       | Make the agent learn a target policy or Minimize the rewards | Yes                                     | No                                  | Yes                                                          | Yes                                                          | Not Given   |
> | LCBT (our work) | Make the agent learn a target policy                         | Yes                                     | No                                  | No                                                           | No                                                           | Given       |
>
> [1] Provably Efficient Black-Box Action Poisoning Attacks Against Reinforcement Learning. NIPS 2021.
>
> [2] Spatiotemporally constrained action space attacks on deep reinforcement learning agents. AAAI 2020.
>
> [3] Vulnerability-aware poisoning mechanism for online rl with unknown dynamics. ICLR 2021.
>
> In the above table, we conducted a comparative analysis of existing literature concerning attack methodologies centered on action manipulation. Notably, our comparison included the metric "The adversary's requisition for neural networks or gradient computations during attack," setting it as one of the distinctions between our approach and other studies. While existing works typically implement attacks from the neural network perspective in continuous action spaces, we consider the neural network as a black box, constructing attacks solely from the theoretical framework of reinforcement learning itself. This distinction forms a key aspect of our current work in contrast to others in the field. Hence, our work did not directly compare with other studies, such as [3], in the experimental setup.
>
> ### "The environments considered are extremely low-dimensional, even by the standards of formal methods research" and "There are only two environments".
>
> We conducted an additional experiment in a five-dimensional space, the environmental setup of which is detailed in Environment 3 within Appendix A. The corresponding experimental outcomes are presented in Figure 7 of Appendix G.
>
> ## Question: Does the analysis hold for stochastic policies? I was a bit unclear on this since the definitions of $Q$ and $V$ both use the notation introduced for deterministic policies. Moreover it seems to me that some parts of the proofs rely on the assumption that all policies are deterministic (including the exploration policies of the RL algorithm), but most RL algorithms do not work by evaluating deterministic policies.
>
> In fact, in this paper, we only require the target policy, i.e., $\pi^{\dagger}$ is a deterministic policy. There is no specific requirement for the reinforcement learning algorithm used by the agent. In addition, we also discuss the impact of the reinforcement learning algorithm (**D-Regret**) used by the agent on our attack cost in the remark of Theorem 2. Also, we carry out experiments to apply the LCBT attack against DDPG, PPO, and TD3. Furthermore, in the proof section, a portion of the proof statements at the beginning of Appendix C (Proof of Theorem 1) could be refined without affecting the subsequent proof (We have performed updates in Appendix C, specifically focusing on the derivation of $\overline{V}^o_1(s^k_1) - \overline{V}^{\pi^k}_1(s^k_1)$).

---

> > ### Comment · Reviewer_hNCZ · 2023-11-21
> >
> > Thank you for the response. The new experiment and the clarification about stochastic policies have addressed my major concerns. In light of these updates I have raised my score (3 -> 6).

---

### Official Review · Reviewer_rSLW · 2023-11-01

**Soundness:** 3 good
**Presentation:** 2 fair
**Contribution:** 2 fair
**Rating:** 5
**Confidence:** 3

**Summary:**

This work investigates the vulnerability of reinforcement learning under action-manipulation attack in the continuous state and action space. In this setting, the goal of the attacker is forcing the learner to learn an approximation of the target policy by altering the learner's action in a continuous action space. The authors study the action-manipulation attack in both white-box and black-box settings. They first propose a white-box oracle attack strategy, which can achieve sublinear attack cost and loss under some assumptions. They propose a black-box attack method named LCBT, which is able to force the RL agent to choose actions according to the policies specified by the attacker with sublinear attack cost. The experimental results show the effectiveness of the proposed attack algorithms.

**Strengths:**

Originality: this is the first work that provides theoretical guarantees on the bound of the attack cost and loss of the action-manipulation attack against RL with continuous state and action spaces. The choice to utilize a binary tree structure is well-founded and effectively addresses the challenges derived from the continuous setting.

Quality: the authors of this paper have developed two attack algorithms for white-box and black-box settings respectively. They offer an in-depth theoretical analysis of these algorithms' attack cost and loss. Additionally, experimental results somewhat support the theoretical results.

Clarity: the main text of this paper conveys the idea and the proposed method well. The theoretical analyses seem to be solid although I do not check the proof in the appendix carefully.

Significance: The authors provide some theoretical guarantees of the proposed attack algorithms which show the attacker can mislead the agent by spending sublinear attack cost and achieve sublinear attack loss. The results are interesting and show that the action attack is also harmful in continuous reinforcement learning.

**Weaknesses:**

1.  If I am not wrong, this paper does not introduce the method of dividing the continuous subset of the action space when new leaves are generated. This part is also important for the practical application of the proposed attack method.

2. The binary tree method can deal with the low-dimensional continuous action space but may meet problems in high-dimensional space. In high-dimensional action space case, the tree method is not efficient and it may requires huge mount of rounds to explore the worst node.

3. The proposed black-box method needs to discretize the continuous state space. This paper partitions it into $M$ subintervals. How can a black-box attacker find a proper discretization of the state space? It seems unrealistic in the high-dimensional state space. As the attack cost and loss are linear dependent on $M$.

**Questions:**

1. What is the method of dividing the continuous subset of the action space when new leaves are generated?

2. Can the method be directly used in high-dimensional action space case?

3. See in Weaknesses 3.

---

> ### Author Response · Authors · 2023-11-20
>
> Thanks for your feedback. I will response to the questions you mentioned.
>
> ## Question 1: What is the method of dividing the continuous subset of the action space when new leaves are generated?
>
> When partitioning a node, it is essential to ensure that the action space corresponding to the child nodes satisfies the condition $diam_a(\mathcal{P}_{D,I})\leq\nu_1\rho^D$, where $D$ represents the depth of the child nodes. In practical applications, the action space is multidimensional, allowing for the configuration of a dimension for each layer in a binary tree. Through iterative correspondence, when a node at a certain layer is partitioned to generate its child nodes, the action space can be halved along the corresponding dimension.
>
> ## Question 2: Can the method be directly used in high-dimensional action space case?
>
> Here, we shall begin by conducting a theoretical analysis of the circumstances surrounding the method proposed by us in high-dimensional settings. One obvious point is that our method requires partitioning of the action and state spaces. With higher dimensions in the action space, even if we use binary tree partitioning, the sufficiency of partitioning will still be lower than in lower dimensions with the same number of partitions. This means that the LCBT algorithm converges slower towards the worst action, resulting in a higher level of non-stationarity in the environment and a larger **D-Regret**. According to Theorem 2, this also leads to higher attack cost.
>
> Additionally, we have introduced a new experiment in a 5-dimensional space, as described in Environment 3 in Appendix A. The corresponding experimental outcomes are presented in Figure 7 of Appendix G. Remarkably, under the conditions of a 5-dimensional space, the LCBT algorithm demonstrates commendable performance. Hence, in practical terms, factors influencing the efficacy of the LCBT algorithm encompass not only its intrinsic parameters such as $\nu$, $\rho$, $d_s$, but also encompass the environment and the algorithms employed by the agent.
>
> ## Question 3:  How can a black-box attacker find a proper discretization of the state space? It seems unrealistic in the high-dimensional state space.
>
> For the partitioning of the state space, theoretically, each substate should satisfy $diam_s(S_i)\leq L_s d_s < \Delta^o_{\dagger}+\Delta^{\dagger}_{min}$. In practical implementation, due to the diversity of the environment, for each subinterval, an attacker only needs to ensure that, for different initial states within the same subinterval and adopting the same policy, the generated Q-values are approximately close, with differences not exceeding the disparity between Q-values produced by employing the target policy and the worst policy for the same initial state (equivalently understood as trajectories being roughly similar, with differences not exceeding the trajectory disparities between employing the target policy and the worst policy for the same initial state). Although Q-values cannot be directly computed, there exists a certain degree of assessability in determining the similarity of trajectories.

---

> > ### Comment · Reviewer_rSLW · 2023-11-21
> >
> > Most of my concerns are addressed. Overall, I like the techniques introduced in this paper. The tree method can somewhat deal with the challenges in the continuous action space. However, the tree method will still meet the curse of dimensionality. As the number of dimensions of features increases, the amount of sample needed accurately increases exponentially. Thus, I will keep my score.

---

### Author Response · Authors · 2023-11-20

1. We conducted an additional experiment in a five-dimensional space, the environmental setup of which is detailed in Environment 3 within Appendix A. The corresponding experimental outcomes are presented in Figure 7 of Appendix G.
2.

| Method          | Attacker's goal                                              | The support for continuous action space | The demand for knowledge of the MDP | The demand for knowledge of the RL algorithm used by the agent | The adversary's requisition for neural networks or gradient computations during attack | Attack cost |
| --------------- | ------------------------------------------------------------ | --------------------------------------- | ----------------------------------- | ------------------------------------------------------------ | ------------------------------------------------------------ | ----------- |
| LCB-H[1]        | Make the agent learn a target policy                         | No                                      | No                                  | No                                                           | No                                                           | Given       |
| LAS[2]          | Minimize the rewards                                         | Yes                                     | Yes                                 | Yes                                                          | Yes                                                          | Not Given   |
| VA2C-P[3]       | Make the agent learn a target policy or Minimize the rewards | Yes                                     | No                                  | Yes                                                          | Yes                                                          | Not Given   |
| LCBT (our work) | Make the agent learn a target policy                         | Yes                                     | No                                  | No                                                           | No                                                           | Given       |

[1] Provably Efficient Black-Box Action Poisoning Attacks Against Reinforcement Learning. NIPS 2021.

[2] Spatiotemporally constrained action space attacks on deep reinforcement learning agents. AAAI 2020.

[3] Vulnerability-aware poisoning mechanism for online rl with unknown dynamics. ICLR 2021.

In the above table, we conducted a comparative analysis of existing literature concerning attack methodologies centered on action manipulation.

---

### Meta-Review · Area_Chair_Hx1U · 2023-12-10

**Metareview:**

This paper proposes novel attack algorithms for manipulating the actions of reinforcement learning agents in continuous state and action spaces. The reviewers raised several valuable points regarding the novelty, theoretical analysis, and experiments, which the authors addressed partly in their responses. The overall scores from the reviewers are very marginal.

Moreover, I wonder why action-manipulation attacks can happen in real-world systems "such as robot control, autonomous driving, game
intelligence, etc.". Taking auto-driving as an example, visual attacks manipulate, e.g., traffic signs, that are outside of the car. But driving actions are inside of the car. If the car does not act correctly, most of the time it is called malfunctioning instead of an attack. Moreover, in many real-world tasks, to deal with issues such as sim2real, a robust policy is trained in randomized environments, which can include dynamics system variations. I doubt the action-manipulation attack can work for robust policies.

**Justification For Why Not Higher Score:**

Besides that the reviewers were not completely satisfied with the response, the scenarios seemed very unreal to me. Particularly when the policy is trained with randomization or adversarial players in many real cases, the study in this paper is useless.

**Justification For Why Not Lower Score:**

N/A

---

### Decision · Program_Chairs · 2024-01-16

Reject